# Mycobacterium tuberculosis Rv2626c-derived peptide as a therapeutic agent for sepsis

Sun Young Kim[1,†], Donggyu Kim[2,3,†], Sojin Kim[2], Daeun Lee[2], Seok-Jun Mun[1,3], Euni Cho[1,3], Wooic Son[2,3], Kiseok Jang[4] & Chul-Su Yang[2,3,*] [iD]

## Abstract

**The Rv2626c protein of *Mycobacterium tuberculosis* is a promising vaccine candidate owing to its strong serum antibody response in patients with tuberculosis. However, there is limited information regarding the intracellular response induced by Rv2626c in macrophages. In this study, we demonstrated that Rv2626c interacts with the RING domain of TRAF6 and inhibits lysine (K) 63-linked polyubiquitination of TRAF6 (E3 ubiquitin ligase activity); this results in the suppression of TLR4 inflammatory signaling in macrophages. Furthermore, we showed that the C-terminal 123–131-amino acid Rv2626c motif promotes macrophage recruitment, phagocytosis, M2 macrophage polarization, and subsequent bacterial clearance. We developed rRv2626c-CA, a conjugated peptide containing the C-terminal 123–131-amino acid Rv2626c that targets macrophages, penetrates the cell membrane, and has demonstrated significant therapeutic effects in a mouse model of cecal ligation and puncture-induced sepsis. This multifunctional rRv2626c-CA has considerably improved potency, with an $IC_{50}$ that is 250-fold (*in vitro*) or 1,000-fold (*in vivo*) lower than that of rRv2626c-WT. We provide evidence for new peptide-based drugs with anti-inflammatory and antibacterial properties for the treatment of sepsis.**

**Keywords** macrophage-targeting; *Mycobacterium tuberculosis* Rv2626c peptide; sepsis; TRAF6

**Subject Category** Microbiology, Virology & Host Pathogen Interaction

## Introduction

Bacterial antigens trigger host immune defense and bacterial immune response, allowing bacteria to evade or protect themselves from host immunity (Hoffmann *et al*, 2018). These host–pathogen interactions are particularly complicated during infection by intracellular pathogens, such as *Mycobacterium tuberculosis* (MTB), the pathogen causing tuberculosis (TB) (Flynn & Chan, 2005). Accumulating evidence supports the importance of MTB antigens in the development of vaccines, diagnostic methods, and therapeutics for infectious diseases, including TB (Coppola & Ottenhoff, 2018; Gago *et al*, 2018; Hoffmann *et al*, 2018; Olive & Sassetti, 2018; Kim *et al*, 2019). MTB Rv2626c is one of the most highly expressed proteins under conditions of hypoxia or nitric oxide-induced stress and can be detected in the culture supernatant and lysates of MTB (Jiang *et al*, 2018) as well as in activated macrophages in the lungs (Shi *et al*, 2003; Bashir *et al*, 2010). Rv2626c has been shown to elicit a strong serum antibody response in patients with active TB, making it a useful biomarker for disease progression (Davidow *et al*, 2005). However, its role remains to be elucidated. Therefore, understanding the dynamic interactions between macrophages and Rv2626c is crucial for developing effective anti-TB strategies.

Sepsis is a life-threatening condition with multiorgan dysfunction caused by a dysregulated host response to infection (van der Poll *et al*, 2017). In sepsis, the immune response initiated by an invading pathogen fails to restore immune homeostasis, thereby manifesting as a pathological syndrome characterized by bacterial growth and excessive inflammation (Kim *et al*, 2016; van der Poll *et al*, 2017; Kim *et al*, 2018). The mortality rate of sepsis is as high as 25%, imposing significant burden on global health (Rhodes *et al*, 2017). Our understanding of the key mechanisms underlying the pathogenesis of sepsis has considerably improved; nevertheless, effective and targeted therapeutic strategies are still warranted. Recent advances in the treatment of sepsis using microbe-derived antigens represent a paradigm shift (Kim *et al*, 2018), making MTB Rv2626c a potentially new therapeutic agent.

Peptides and proteins have great potential as therapeutics. Compared with the currently popular small-molecule drugs, peptides and proteins are more specific because they have multiple points of contact with their targets (Bruno *et al*, 2013; Craik *et al*, 2013). In addition, increased specificity results in decreased side effects and toxicity. Peptides can be designed to target a broad range of molecules, providing almost limitless possibilities of applications in the fields of oncology, immunology, infectious disease, and endocrinology (Bruno *et al*, 2013; Koh *et al*, 2017; Kim *et al*, 2018; Kim *et al*, 2020). In this study,

1 Department of Bionano Technology, Hanyang University, Seoul, South Korea
2 Department of Molecular and Life Science, Hanyang University, Ansan, South Korea
3 Center for Bionano Intelligence Education and Research, Hanyang University, Ansan, South Korea
4 Department of Pathology, Hanyang University College of Medicine, Seoul, South Korea
*Corresponding author. Tel: +82 31 400 5519; Fax: +82 31 436 8153; E-mail: chulsuyang@hanyang.ac.kr
†These authors contributed equally to this work

we revealed the minimal essential peptide motif of Rv2626c with therapeutic efficacy against lethal inflammatory diseases.

Despite continuous advances in the development of therapeutic peptides and proteins, methods to increase systemic stability and site-specific delivery are lacking. Furthermore, the poor target specificity of peptides remains a major obstacle (Bruno et al, 2013; Craik et al, 2013; Jain et al, 2013). To address this issue, we conjugated Rv2626c peptides to macrophage-targeting, cell-penetrating peptides, thereby improving their efficacy for the treatment of sepsis.

In this study, we identified and quantified an Rv2626c interaction complex using mass spectrometry-based proteomic analysis of MTB-infected macrophages. We demonstrated that the C-terminal 9-amino acid Rv2626c peptide interacts with TRAF6 and blocks K63-linked polyubiquitination (E3 ubiquitin ligase activity), thereby suppressing inflammation and enhancing antibacterial activity. Subsequently, treatment with macrophage-targeting rRv2626c-CA dramatically reduced the mortality of mice with cecal ligation and puncture (CLP)-induced polymicrobial sepsis. These results suggest that Rv2626c peptide could be a powerful tool in therapeutic interventions against sepsis.

# Results

### rRv2626c regulates inflammatory response in macrophages via the TLR signaling pathway

To determine the role of Rv2626c in the innate immune response of macrophages, we purified recombinant His-tagged rRv2626c, as described previously (Koh et al, 2017; Kim et al, 2020). The purified rRv2626c (16 and 36 kDa) was validated via sodium dodecyl sulfate–polyacrylamide gel electrophoresis and immunoblotting (Fig 1A and Appendix Fig S1A). No significant rRv2626c-induced cell cytotoxicity was observed in macrophages (Fig 1B). Furthermore, immunostaining and cell fractionation revealed that rRv2626c was distributed in the cytoplasm of macrophages (Fig 1C and Appendix Fig S1B).

Previous studies have shown that rRv2626c binds to the cell surface and localizes inside macrophages, thereby regulating the NF-κB pathway and promoting cytokine production (Shi et al, 2003; Bashir et al, 2010; Danelishvili et al, 2016). To characterize the function of rRv2626c in macrophages, we measured rRv2626c-induced cytokine production using ELISA. As shown in Fig 1D and E, rRv2626c promoted the production of proinflammatory cytokines (TNF-α, IL-6, and IL-12p40) and anti-inflammatory cytokines (IL-10) via the TLR2/MyD88/TRAF6/IRAK1 pathway, instead of the TRIF-dependent (TBK1) pathway. In addition, we observed the internalization of rRv2626c by macrophages. Interestingly, His-tagged rRv2626c was not detected in cell lysates when expressed in TLR2$^{-/-}$ primary bone marrow-derived macrophages (BMDMs) (Fig 1F). Taken together, these results suggest that rRv2626c is internalized into the macrophages via TLR2 and that the TLR2/MyD88-dependent pathway is crucial for the rRv2626c-mediated inflammatory response.

### rRv2626c negatively regulates TLR-mediated inflammatory responses in macrophages

Next, to investigate the role of Rv2626c in the TLR-mediated inflammatory signaling pathway (Bashir et al, 2010; Sun et al,

2017), we examined whether Rv2626c inhibits inflammatory cytokine production induced by TLR4 (LPS) or TLR2/6 (Pam$_2$CSK$_4$). As shown in Fig 2A and B, the TLR-mediated production of TNF-α, IL-6, and IL-12p40 (proinflammatory cytokines) was attenuated by rRv2626c, whereas that of IL-10 (an anti-inflammatory cytokine) was promoted by rRv2626c. Therefore, we hypothesized that rRv2626c enters macrophages via TLR2 and that intracellular rRv2626c regulates the TLR signaling pathway. Because the PI3K, MAPK, and NF-κB signaling pathways were involved in inflammatory signaling in LPS-challenged macrophages (Kim et al, 2016), we investigated whether the activation of the signaling proteins in these pathways was affected by rRv2626c. Interestingly, rRv2626c pretreatment inhibited LPS-induced phosphorylation of AKT, MAPK, and IκBα and degradation of IκBα (Fig 2C). Taken together, these results suggest that rRv2626c acts as a negative regulator to suppress TLR ligand-induced activation of the PI3K, MAPK, and NF-κB signaling pathways, thereby inhibiting proinflammatory cytokine production and promoting anti-inflammatory cytokine production in macrophages.

### Rv2626c interacts with TRAF6

We hypothesized that rRv2626c regulates the LPS-induced TLR signaling pathway by interacting with signaling molecules during innate immune response. To identify the binding partners of Rv2626c, we transfected THP-1 cells with His-tagged rVector or rRv2626c and performed coimmunoprecipitation using His-agarose beads. Mass spectrometry revealed several proteins that bind to rRv2626c, including TLR2 (100 kDa), IRAK1 (80 kDa), RIP1 (70 kDa), TRAF6 (60 kDa), MAPK11 (41 kDa), and TOLLIP (32 kDa) (Fig 3A). In vivo analysis of HEK293T cells and macrophages showed that Rv2626c strongly interacts, although temporarily (30–60 min), with endogenous TRAF6 (Fig 3B and C). rRv2626c and TRAF6 immunostaining and confocal microscopy revealed that rRv2626c colocalizes with endogenous TRAF6, mainly in the cytoplasm of macrophages (Fig 3D). These results suggest that rRv2626c directly interacts with TRAF6, a signaling molecule in the TLR pathway.

### CBS2 domain of Rv2626c interacts with the N-terminus of TRAF6 and attenuates its E3 ubiquitin ligase activity

In HEK293T cells, we found that the CBS2 domain of Rv2626c is required for its interaction with TRAF6 (Fig 4A) (Shi et al, 2003). To further characterize the minimal peptide sequence of Rv2626c that mediates its interaction with TRAF6, we designed 9–10-amino acid fragments of the CBS2 domain and fused it with the transduction domain of the HIV-1 Tat protein to form a retro-inverso peptide (Tat-CBS2 peptide) for intracellular delivery in order to prevent proteolytic breakdown (Kim et al, 2016; Kim et al, 2020; Tables 1 and 2). These peptides were tested for the minimal residues required for TRAF6 interaction in HEK293T cells. Our results showed that the Tat-CBS2 ($_{123}$LPEHAIVQF$_{131}$) peptide was capable of blocking Rv2626c interaction with TRAF6, whereas the Tat peptide alone was unable to do so (Fig 4B). Notably, $_{125}$EH$_{126}$ amino acids with charged side chains in the peptide ($_{123}$LPEHAIVQF$_{131}$) were essential for its interaction with TRAF6. Furthermore, we examined various truncated mutants of TRAF6 and showed that the RING domain of TRAF6 is required for its

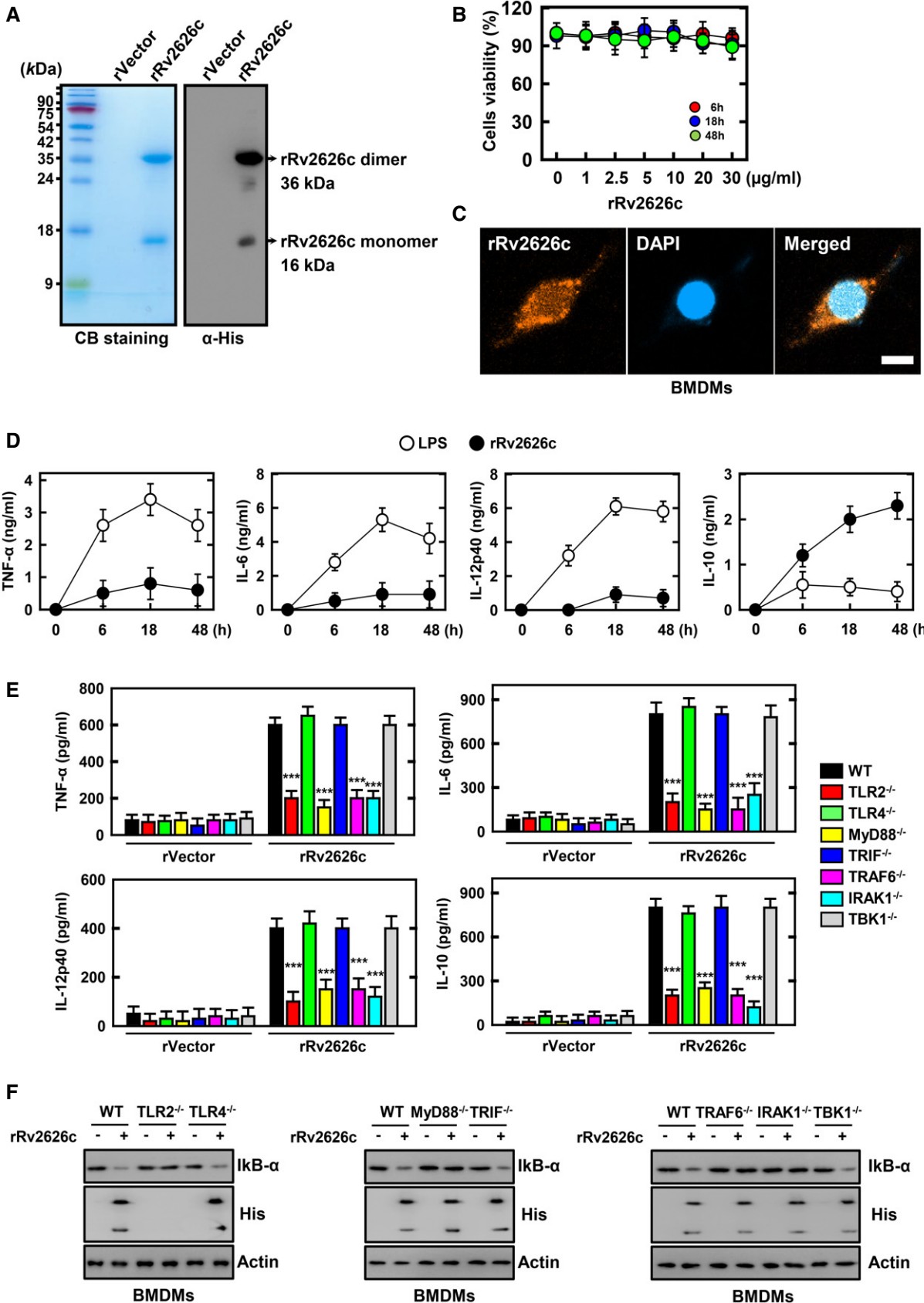

**Figure 1.**

◄

**Figure 1.  Effects of rRv2626c on inflammatory response in macrophages.**

A       Bacterially purified 6xHis-rRv2626c and analyzed by Coomassie blue staining (left) or immunoblotting (IB) with αHis (right).

B       BMDMs were incubated with rRv2626c for the indicated times and then cell viability measured with MTT assay.

C       Fluorescence confocal images showing in BMDMs treated with Rv2626c (2.5 μg/ml) for 30 min, fixed, immunostained with antibodies for His (Alexa 488) and DAPI. Scale bar, 10 μm.

D, E    BMDMs from WT mice were treated with LPS (100 ng/ml) or rRv2626c for indicated times (D). BMDMs from WT, TLR2$^{-/-}$, TLR4$^{-/-}$, MyD88$^{-/-}$, TRIF$^{-/-}$, IRAK1$^{-/-}$, TRAF6$^{-/-}$, and TBK1$^{-/-}$ mice were treated with 2.5 μg/ml rRv2626c or rVector for 18 h (E). Culture supernatants were harvested, and the levels of TNF-α, IL-6, IL-12p40, and IL-10 were measured by ELISA.

F       BMDMs were treated with 2.5 μg/ml rRv2626c for 1 h, harvested, and then subjected to IB for αIκBα and His. α-Actin was used as a loading control.

Data information: Data shown are representative of three independent experiments with similar results (A, C, E). Data shown are the means ± SD of five experiments (B, D, E). Statistical significance was determined by Student's *t*-test with Bonferroni adjustment (***$P < 0.001$) compared with rVector.

interaction with Rv2626c (Fig 4C). Taken together, these results indicate that Rv2626c directly interacts with the TRAF6 N-terminal RING domain via its CBS2 domain, particularly the CBS2 (aa 123–131) motif (Appendix Fig S2A).

Because TRAF6 polyubiquitination (E3 ubiquitin ligase activity) is a key regulatory event in TRAF6 activation and downstream NF-κB signaling pathway (Wang *et al*, 2001; Yang *et al*, 2016), we investigated whether the interaction between Rv2626c and TRAF6 affects TRAF ubiquitination. Western blotting revealed that the expression of HA-tagged ubiquitin led to the polyubiquitination of a Flag-tagged TRAF6; this ubiquitination was attenuated by Rv2626c expression in a dose-dependent manner (Fig 4D). To further determine the type of ubiquitin chain regulated by Rv2626c, we used HA-tagged K48-linked and K63-linked ubiquitin plasmids. It is well-known that K48-linked ubiquitination mediates the proteasomal degradation of ubiquitinated substrates, whereas K63-linked ubiquitination is essential for the activation of NF-κB and downstream signaling pathways (Wang *et al*, 2001; Yang *et al*, 2016). We showed that the overexpression of K48-linked and K63-linked ubiquitin chain both induced TRAF6 ubiquitination (Fig 4D). The presence of Rv2626c had no effect on K48-linked ubiquitin chain, whereas K63-linked ubiquitination was strongly inhibited by Rv2626c expression in a dose-dependent manner (Fig 4D and E, and Appendix Fig S2B). Our results indicate that Rv2626c binds to TRAF6 and inhibits K63-linked polyubiquitination of TRAF6 (E3 ubiquitin ligase activity), which activates NF-κB and its downstream signaling pathway.

**Tuftsin-conjugated Rv2626c peptide targets macrophages**

Tuftsin is a natural tetrapeptide (Thr/Lys/Pro/Arg) derived from the proteolytic cleavage of 289–292 amino acids of IgG in the spleen. It modulates immune response by stimulating phagocytosis (Wang *et al*, 2001; Liu *et al*, 2012; Wu *et al*, 2012). We designed the Tuftsin-conjugated 123–131-amino acid Rv2626c peptide that potentially targets macrophages, penetrates the cell membrane, and regulates immune responses (Fig 5A). To assess the function of Rv2626c-CA (constitutive active form) and Rv2626c-DN (dominant-negative form, E125Q and H126Q) peptides in macrophages, we expressed His-tagged rRv2626c-CA and its dominant-negative mutant in bacteria and purified them via affinity chromatography, as previously described (Koh *et al*, 2017; Kim *et al*, 2020). The purified rRv2626c-CA and its mutant (10 kDa) were validated via sodium dodecyl sulfate–polyacrylamide gel electrophoresis and

immunoblotting (Fig 5B). No significant differences in macrophage cytotoxicity were observed for rRv2626c-CA and its mutant compared with the vehicle controls in BMDMs using the MTT assay (Fig 5C).

We verified whether rRv2626c-CA mimics the pharmacological and biological profiles of wild-type rRv2626c (rRv2626c-WT) (Figs 2 and 3). Consistent with the activity of rRv2626c-WT, rRv2626c-CA also regulates inflammatory responses in macrophages in a dose-dependent manner. Remarkably, rRv2626c-CA had an IC$_{50}$ value of 10 ng/ml, which was 250-fold lower than that (2.5 μg/ml) of rRv2626c-WT (Fig 5D). Additionally, rRv2626c-CA treatment dramatically suppressed the LPS-induced interaction between endogenous TRAF6 and Rv2626c and TRAF6 (Fig 5E). Notably, no significant changes in inflammatory response or TRAF6 binding and its E3 ubiquitin ligase activity were observed upon rRv2626c-DN (lost TRAF6 binding) or rVehicle (tuftsin) treatment of macrophages (Fig 5D and E, and Appendix Fig S3). Therefore, rRv2626c-CA acts as a specific and potent inflammatory modulator of the macrophage-mediated immune response.

**rRv2626c-CA protects mice from systemic sepsis**

We further investigated whether rRv2626c-CA protects mice from CLP-induced polymicrobial sepsis. First, we tested the therapeutic efficacy of rRv2626c-CA against CLP-induced polymicrobial sepsis in mice. Intraperitoneal injection of rRv2626c-CA into CLP-treated mice at four time points (0, 6, 12, and 18 h) resulted in a dose-dependent protection; 90% of the mice were protected from CLP-induced mortality after rRv2626c-CA administration at a dose of 100 μg/kg per mouse. Remarkably, rRv2626c-CA had an IC$_{50}$ value of 20 μg/kg, which was 1,000-fold more potent than that of rRv2626c-WT (IC$_{50}$: 20 mg/kg; Fig 6A and B). No significant change in survival curves was observed between rRv2626c-DN (lost TRAF6 binding) and rVehicle (tuftsin) treatment (Fig 6C). Consistent with the mortality data, serum concentrations of the proinflammatory cytokines including TNF-a, IL-6, IL-1β, and IL-12p40 significantly reduced in mice treated with rRv2626c-CA. However, the concentration of IL-10 significantly increased following rRv2626c-CA treatment (Fig 6D). These changes were accompanied by increased hematoxylin and eosin staining, suggesting the decreased infiltration of immune cells and decreased damage to the lung, liver, and spleen (Fig 6E). Next, we tested whether rRv2626c-CA has similar activity *in vivo*, which was determined by *in vivo* TRAF6 interaction and ubiquitination. Consistent with

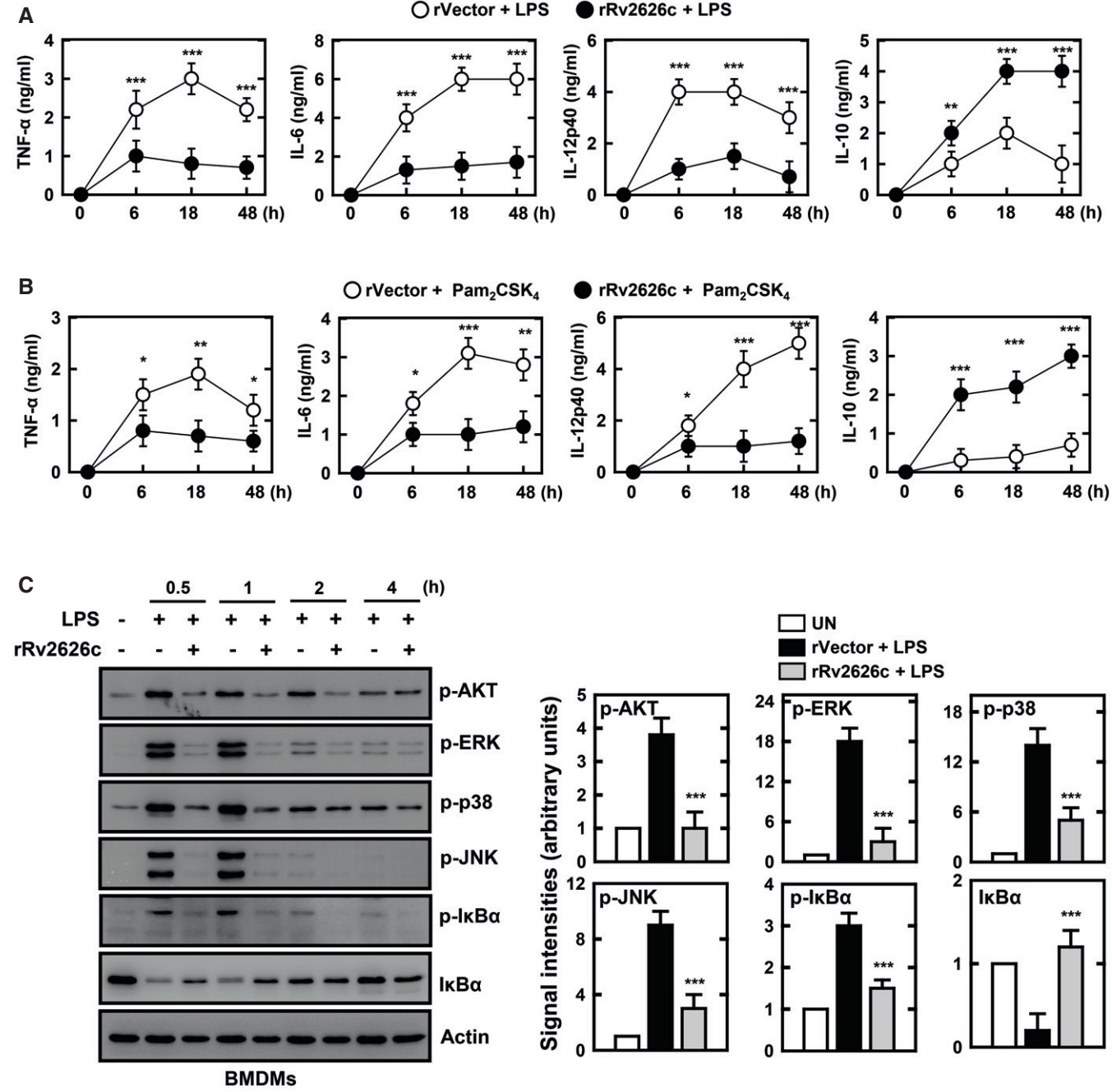

**Figure 2. TLR-mediated inflammatory responses in macrophages are negatively regulated by rRv2626c.**

A–C   BMDMs were pretreated with 2.5 μg/ml rRv2626c or rVector for 1 h and stimulated with 100 ng/ml LPS (A and C) or Pam₂CSK₄ (B) for indicated times. (A and B) Culture supernatants were harvested, and the levels of TNF-α, IL-6, IL-12p40, and IL-10 were measured by ELISA. (C) Cells were harvested and then subjected to IB for the phosphorylated forms of AKT, MAPK (ERK, p38, JNK), IκBα, total forms of IκBα. α-Actin was used as a loading control (left). The densitometric values were calculated from the ratios of protein levels relative to actin (right). Data shown are the means ± SD of three experiments (A and B). Data shown are representative of five independent experiments with similar results (C). Statistical significance was determined by Student's t-test with Bonferroni adjustment (*$P < 0.05$; **$P < 0.01$; ***$P < 0.001$) compared with rVector.

the *in vitro* data (Fig 4E and Appendix Fig S2B), rRv2626c-CA treatment markedly decreased its interaction with TRAF6 and its E3 ubiquitin ligase activity in splenocytes (Fig 6F). Figure 6G shows the pharmacokinetics of rRv2626c-CA distribution in mice after the administration of fluorescent rRv2626c-CA (rRv2626c-CA/Cy5.5). rRv2626c-CA/Cy5.5 was retained in the body for 72 h. It reached its peak concentration in the liver at 18 and 24 h. Furthermore, therapeutic rRv2626c-CA was specifically localized

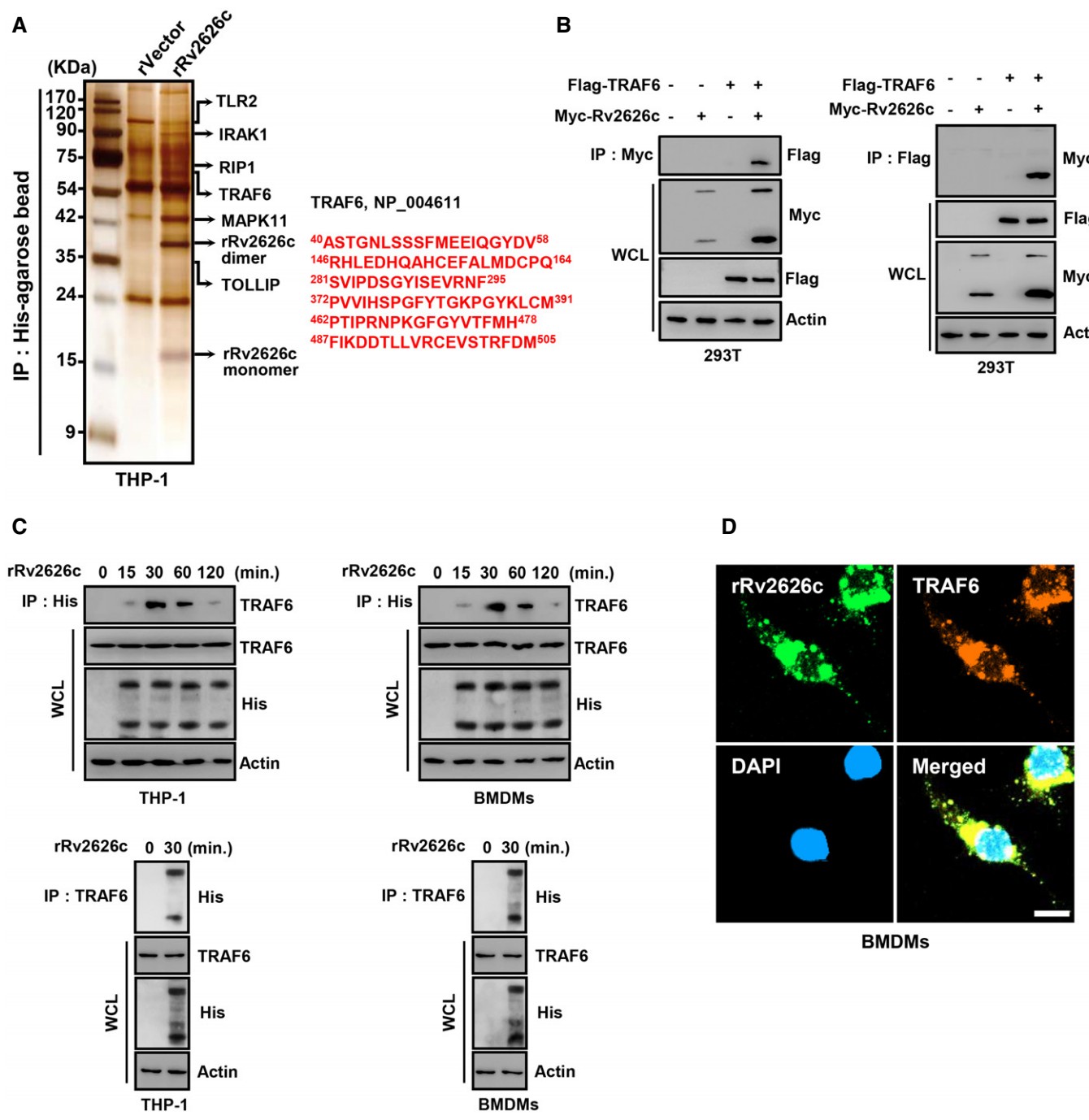

**Figure 3. TRAF6 interacts with Rv2626c.**

A  Identification of TLR2, IRAK1, RIP1, TRAF6, MAPK11, and TOLLIP as endogenous binding partners of rRv2626c. THP-1 cell lysates were incubated with His-tagged rRv2626c or rVector and then IP with His-agarose beads. Binding partners were confirmed by silver staining and mass spectrometric analysis. The red-colored letters indicate the peptides identified from mass spectrometry analysis.

B  293T cells were cotransfected with Flag-TRAF6 and Myc-Rv2626c and IP with αFlag or αMyc. WCLs were used for IB with αFlag, αMyc, or α-actin.

C  THP-1 and BMDMs were stimulated with rRv2626c (2.5 μg/ml) for the indicated times, followed by IP with αHis (up) or αTRAF6 (down) and IB with αTRAF6, αHis, and α-actin.

D  Immunostaining of BMDMs treated with 2.5 μg/ml rRv2626c for 30 min and then immunolabeled with antibody to His-rRv2626c (Alexa Fluor 488) or TRAF6 (Alexa Fluor 568). DAPI (blue) stained nuclei. Scale bar, 10 μm. The data are representative of five independent experiments with similar results (A–D).

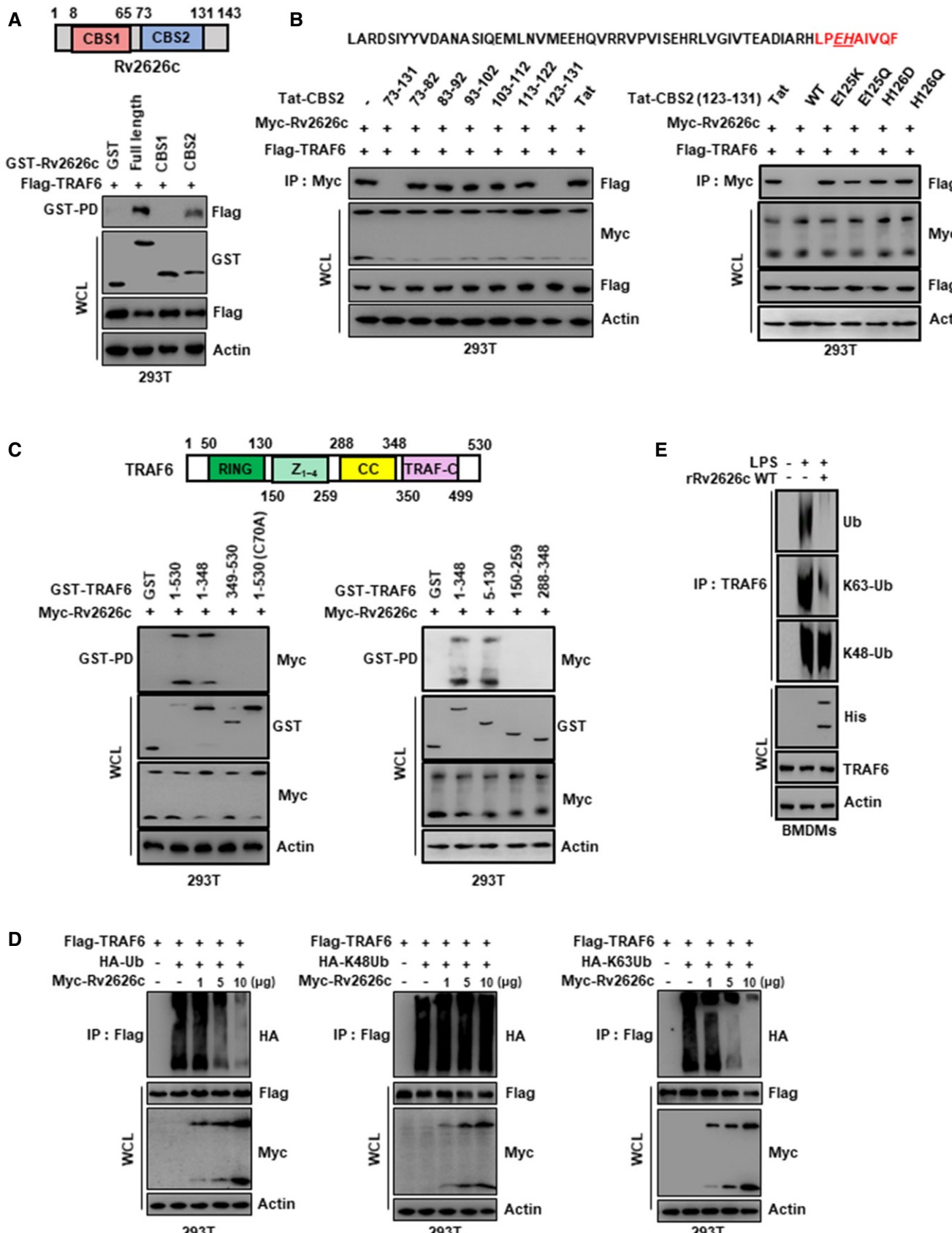

**Figure 4.**

**Figure 4. Rv2626c associates with N-terminal region of TRAF6 through CBS2 domain and attenuates E3 ubiquitin ligase activity of TRAF6.**

- A  Schematic diagram of the structures of Rv2626c (up). 293T cells were cotransfected with Flag-TRAF6 together with GST-vector or GST-Rv2626c and its mutant constructs, subjected to GST pulldown, followed by IB with αFlag. WCLs were used for IB with αGST, αFlag, or α-actin (down).
- B  At 12 h post-transfection with mammalian Myc-Rv2626c constructs together with Flag-TRAF6 and 293T cells treated with several Tat-CBS2 peptide for 12 h (10 μM) for 6 h, subjected to IP with αMyc, followed by IB with αFlag. WCLs were used for IB with αMyc, αFlag, or α-actin.
- C  Schematic diagram of the structures of TRAF6 (up). 293T cells were cotransfected with Myc-Rv2626c together with GST-vector or GST-TRAF6 and its mutant constructs, subjected to GST pulldown, followed by IB with αMyc. WCLs were used for IB with αGST, αMyc, or α-actin (down).
- D  293T cells were cotransfected with Flag-TRAF6, Myc-Rv2626c as indicated doses and HA-ubiquitin (left), HA-K48-linked ubiquitin (middle), HA-K63-linked ubiquitin (right) and then IP with αFlag, followed by IB with αHA. WCLs were used for IB with αFlag, αMyc, or α-actin.
- E  BMDMs were pretreated with 2.5 μg/ml rRv2626c for 1 h and stimulated with 100 ng/ml LPS for 30 min, followed by IP with αTRAF6, IB with ubiquitin, K48-linked ubiquitin, or K63-linked ubiquitin. WCLs were used for IB with αHis, αTRAF6, or α-actin. The data are representative of five independent experiments with similar results (A–E).

**Table 1. Amino acid sequences of *Mycobacterium tuberculosis* Rv2626c.**

| Gene | Accession no | Sequence |
|------|-------------|----------|
| NC_000962.3 | NP_217142.1 | MTTARDIMNAGVTCVGEHETLTAAAQYMREHDIGALPICGDDDRLHGMLTDRDIVIKGLAAGLDPNTATAGELARDSIYYVDANASIQEMLNVMEEHQVRRVPVISEHRLVGIVTEADIARHLPEHAIVQFVKAICSPMALAS |

Pink Box: CBS1 domain of Rv2626c (aa 8-65). Blue Box: CBS2 domain of Rv2626c (aa 73-131).

**Table 2. Sequences of *Mycobacterium tuberculosis* Rv2626c peptide.**

| Name | Sequence (From N to C) |
|------|------------------------|
| Tat | RRRQRRKKRGY |
| Tat-Rv2626c-(73–131) | RRRQRRKKRGY-G-LARDSIYYVDANASIQEMLNVMEEHQVRRV PVISEHRLVGIVT EADIARHLPEHAIVQF |
| Tat-Rv2626c-(73–82) | RRRQRRKKRGY-G-LARDSIYYVD |
| Tat-Rv2626c-(83–92) | RRRQRRKKRGY-G-ANASIQEMLN |
| Tat-Rv2626c-(93–102) | RRRQRRKKRGY-G-VMEEHQVRRV |
| Tat-Rv2626c-(103–112) | RRRQRRKKRGY-G-PVISEHRLVG |
| Tat-Rv2626c-(113–122) | RRRQRRKKRGY-G-IVTEADIARH |
| Tat-Rv2626c-(123–131) | RRRQRRKKRGY-G-LPEHAIVQF |
| Tat-Rv2626c-(E125K) | RRRQRRKKRGY-G-LPKHAIVQF |
| Tat-Rv2626c-(E125Q) | RRRQRRKKRGY-G-LPQHAIVQF |
| Tat-Rv2626c-(H126D) | RRRQRRKKRGY-G-LPEDAIVQF |
| Tat-Rv2626c-(H126Q) | RRRQRRKKRGY-G-LPEQAIVQF |

in macrophages, instead of T and B cells, in the lung, liver, and spleen (Fig 6H and Appendix Fig S4). Taken together, these findings suggest that rRv2626c-CA is stable *in vivo*, circulates in the body for a significant amount of time, and specifically targets macrophages, indicating that it is an excellent therapeutic agent for treating CLP-induced polymicrobial sepsis.

### rRv2626c-CA promotes bacterial clearance

CLP-induced mortality positively correlates with bacterial colony counts in the peripheral blood and peritoneal fluid (Lee *et al*, 2015). Intraperitoneal injection of rRv2626c-CA into CLP mice dramatically decreased bacterial colony counts in both the peripheral blood and peritoneal fluid (Fig 7A). However, a direct killing effect of rRv2626c-CA on bacteria was not observed (Appendix Fig S5A). Because bactericidal effects in polymicrobial sepsis have been reported to be mainly mediated by neutrophils (Bashir *et al*, 2016), we examined whether rRv2626c-CA increases immune cell recruitment in an CLP-induced polymicrobial sepsis model. rRv2626c-CA treatment significantly increased the number of macrophages recruited to the spleen and lung, whereas rVehicle and rRv2626c-DN treatments did not (Fig 7B and Appendix Fig S5B). Notably, macrophages recruited by rRv2626c-CA comprised increased M2 macrophages and decreased M1 macrophages (Fig 7C). To determine how rRv2626c-CA contributes to macrophage recruitment, we examined the expressions of phagocytic receptor (scavenger receptor A and Fc receptor), innate immunity receptors (TLR2 and TLR4), and neutrophil receptor (neuropilin-1 and CXCR2) in macrophages. As shown in Fig 7D, rRv2626c-CA promoted the expressions of scavenger receptor A and Fc receptor in macrophages. These results suggest that rRv2626c-CA induces antibacterial effects via M2 macrophage polarization and recruitment as well as enhances phagocytosis.

## Discussion

In this study, we identified a new antiseptic therapeutic approach using an Rv2626c peptide, which penetrates the cell membrane, targets macrophages, and induces anti-inflammatory and antibacterial immune responses. Our findings represent a potential paradigm shift in the development of an urgently warranted therapeutic intervention. In summary, we found that (i) the C-terminal 123–131-amino acid Rv2626c motif containing two charged residues is required for its interaction with TRAF6; (ii) Rv2626c interacts with the RING domain of TRAF6 and inhibits K63-linked polyubiquitination of TRAF6 (E3 ubiquitin ligase activity), thereby inhibiting TLR4 inflammatory signaling in macrophages; (iii) the Tuftsin-conjugated Rv2626c ($_{123}$LPEHAIVQF$_{131}$) protein is targeted to macrophages *in vitro* and *in vivo*; (iv) rRv2626c-CA shows considerably improved potency and specificity compared with rRv2626c-WT; (v) rRv2626c-CA shows more efficient bacterial clearance via M2 macrophage polarization and recruitment as well as enhances phagocytosis; and (vi) rRv2626c-CA protects mice with polymicrobial infection from sepsis. These findings offer a new therapeutic tool for treating septic shock and other microbe-mediated diseases and for reducing mortality in humans. Our results provide new insights into the function of the Rv2626c-derived peptide in targeting macrophages and inhibiting inflammation and bacterial infection.

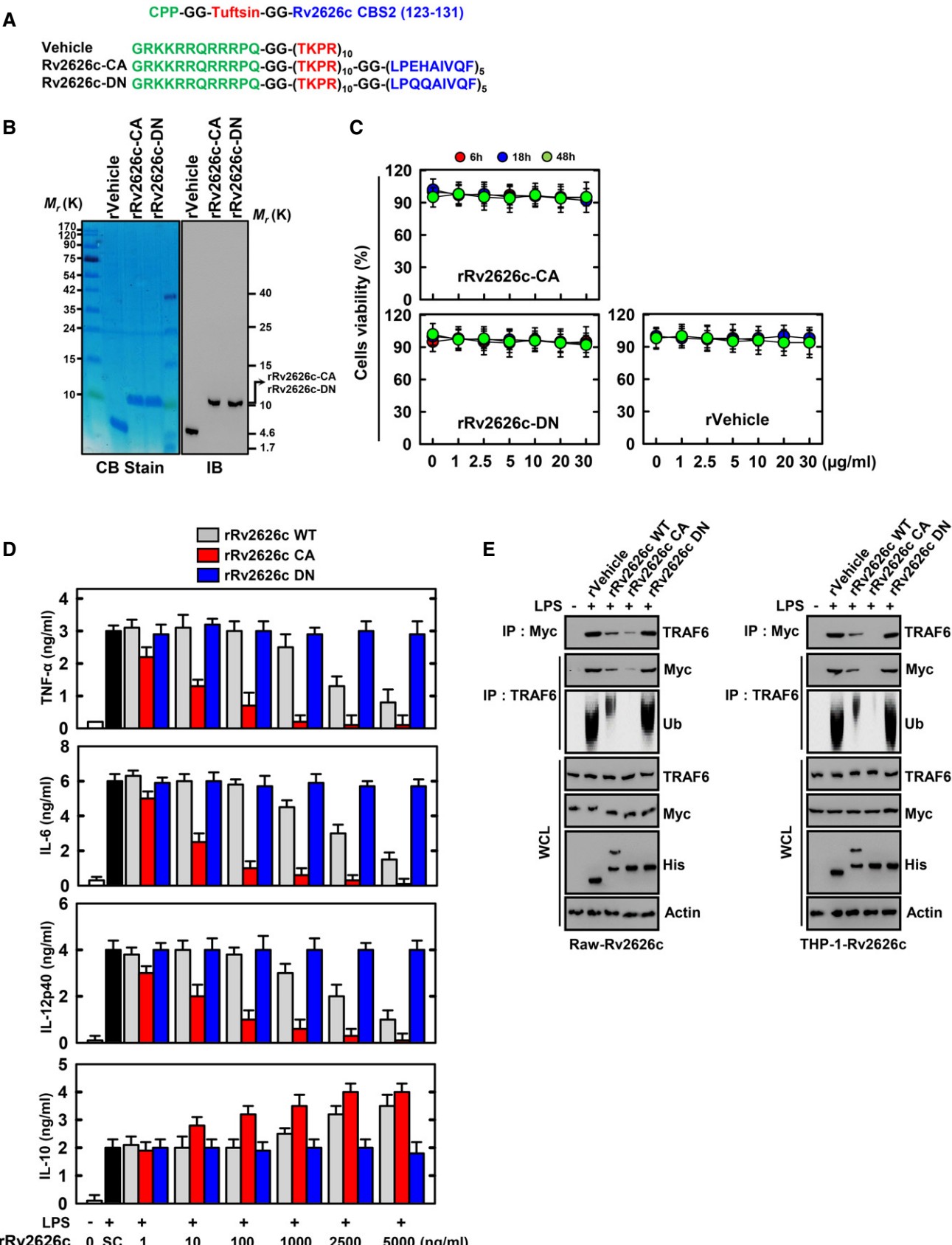

**Figure 5.**

◀

**Figure 5.   Design and expression of Tuftsin-conjugated Rv2626c peptide-based protein.**

A   Schematic in design of rRv2626c and its mutants.

B   Bacterially purified 6xHis-rRv2626c and analyzed by Coomassie blue staining (left) or immunoblotting (IB) with αHis (right).

C   BMDMs were incubated with rRv2626c-CA, rRv2626c-DN, or rVehicle for the indicated times and then cell viability measured with MTT assay.

D   BMDMs were pretreated with rRv2626c-WT, rRv2626c-CA, or rRv2626c-DN for 1 h and stimulated with 100 ng/ml LPS for 18 h. Culture supernatants were harvested, and the levels of TNF-α, IL-6, IL-12p40, and IL-10 were measured by ELISA.

E   Myc-Rv2626c-expressing Raw264.7 or THP-1 cells were pretreated with rRv2626c-WT (2.5 μg/ml), rRv2626c-CA (10 ng/ml), rRv2626c-DN (2.5 μg/ml), for 1 h, and stimulated with 100 ng/ml LPS for 30 min, followed by IP with αMyc or αTRAF6, IB with αMyc and ubiquitin. WCLs were used for IB with αMyc, αTRAF6, αHis, or α-actin. The data are representative of five independent experiments with similar results (B and E). Data shown are the means ± SD of three experiments (C and D). SC, solvent control (PBS).

Emerging evidence suggests that MTB Rv2626c plays a critical role in stimulating macrophages to provoke a an proinflammatory response and necrosis, thereby promoting bacterial escape and mycobacterial survival during infection (Danelishvili *et al*, 2016; Sun *et al*, 2017), Furthermore, Rv2626c activates the TH1 response and inhibits regulatory T cells in the peripheral blood of patients with tuberculosis (Singh *et al*, 2019). Therefore, Rv2626c is important for understanding the role of dormancy-related proteins in the latent state of tuberculosis as well as for the discovery of intracellular regulatory mechanisms involved in Rv2626c-induced host response in macrophages. We found that the C-terminal 123–131-amino acid Rv2626c motif interacts with TRAF6, which plays a crucial role in inflammatory responses. (Chen, 2012) demonstrated that Rv2626c contains the RING domain that confers E3 ligase activity in its autoubiquitination via K63 polyubiquitination, which inhibits the inflammatory responses mediated by the NF-κB and MAPK signaling pathways. Consistent with these findings, Rv2626c inhibited the K63 polyubiquitination of TRAF6 and inflammation.

A previous report suggested that the MTB PPE18/Rv1196 protein also reduces inflammation and increases survival in an animal model of sepsis (Ahmed *et al*, 2018). Similar to rRv2626c, PPE18 also binds to TLR2. However, PPE18 selectively downregulates proinflammatory immune responses, i.e., the phosphorylation of suppressor of cytokine signaling 3. This results in the physical interaction of PPE18 with the IκBα-NF-κB/rel complex, thereby preventing the phosphorylation and degradation of IκBα and nuclear translocation of p50 and p65 NF-κB and c-rel transcription factors. As a result, the transcription of NF-κB-regulated genes, such as IL-12 and TNF-α, is downregulated (Nair *et al*, 2011). Furthermore, PPE18 increases the secretion of an anti-inflammatory cytokine (IL-10) via the activation of p38 MAPK and M2 macrophages in

macrophages (Nair *et al*, 2009; Ahmed *et al*, 2018). Unlike rPPE18 (which contains 391 amino acids), rRv2626c (which contains nine amino acids) has the following advantages: mechanistically rational design, intracellular delivery of proteins and stability in the blood, and can reduce unexpected off-target side effects in animal experiments. Compared with the concentrations used in animal experiments, rRv2626c has considerably improved potency, with a 500-fold (*in vivo*) lower value than that of rPPE18 (5 mg/kg). Furthermore, Rv2626c may function like antibiotics by improving bacterial clearance via M2 macrophage polarization and recruitment as well as enhance phagocytosis in sepsis. Therefore, the ability of the C-terminal 123–131-amino acid Rv2626c motif to dampen harmful factors and promote protective response in sepsis makes rRv2626c-CA a promising therapeutic agent for treating sepsis.

There is an increase in research focused on cell-targeting peptides. Tuftsin provides a strategy for specifically delivering therapeutic agents to macrophages. The primary function of tuftsin or tuftsin-like peptides is to enhance phagocyte respiratory burst, migration/chemotaxis ability, and antigen presentation in cells of monocytic origin, including macrophages, neutrophils, microglia, and Kupffer cells, thereby increasing antimicrobial and antitumor activities (Liu *et al*, 2012; Wu *et al*, 2012; Gao *et al*, 2016). In animal models, tuftsin or tuftsin-like peptides have been found to improve innate immunity and survival (Wardowska *et al*, 2009; Wu *et al*, 2012; Gao *et al*, 2016).

In this study, we generated rTuftsin (rVehicle) and rTuftsin-conjugated Rv2626c proteins, which are more cell-specific, unlike previous studies that used tuftsin peptides to improve the stability and targeting function of macrophages of the peptide (Appendix Fig S4). rTuftsin lost the previously reported function of controlling the inflammation and antibacterial activity of tuftsin

**Figure 6.   The rRv2626c-CA protects mice from cecal ligation and puncture (CLP)-induced polymicrobial sepsis.**

A–C   Schematic of the CLP model treated with rRv2626c-CA or its mutants (upper). The survival of mice was monitored for 10 days; mortality was measured for *n* = 25 mice per group (lower). Statistical differences compared with the rVector-treated mice are indicated (log-rank test). The data are representative of two independent experiments with similar results.

D, E   (D) Serum cytokine levels and (E) representative hematoxylin and eosin (H&E) staining of the lung, liver, and spleen (left) from 10 mice per group. Histopathology scores were obtained from H&E stained as described in Materials and Methods (right) were determined at 20 h in CLP mice were treated with rRv2626c-CA or its mutants. Scale bar, 100 μm.

F   Splenocytes were used for IP with αHis or αTRAF6, followed by IB with αHis or αUb. WCLs were used for IB with αTRAF6, αHis or α-actin.

G   *In vivo* imaging using IVIS spectrum-chromatography CT system. Intraperitoneal administration of rRv2626c-CA/Cy5.5 was carried out at intervals of 0, 6, 12, and 18 h in a CLP-induced sepsis model. Pharmacokinetics and biodistribution were observed in spleen, liver, lung, and kidney tissues. The data are representative of three independent experiments with similar results.

H   The percentage of F4/80+ macrophages cells and rRv2626c-CA-His were found in the spleen, liver, and lung using Fluorescence-activated cell sorting analysis in the background of CLP-induced sepsis. The data are representative of five independent experiments with similar results (E–H). Statistical significance was determined by Student's *t*-test with Bonferroni adjustment (\*\*\**P* < 0.001) compared with CLP+ rVehicle.

▶

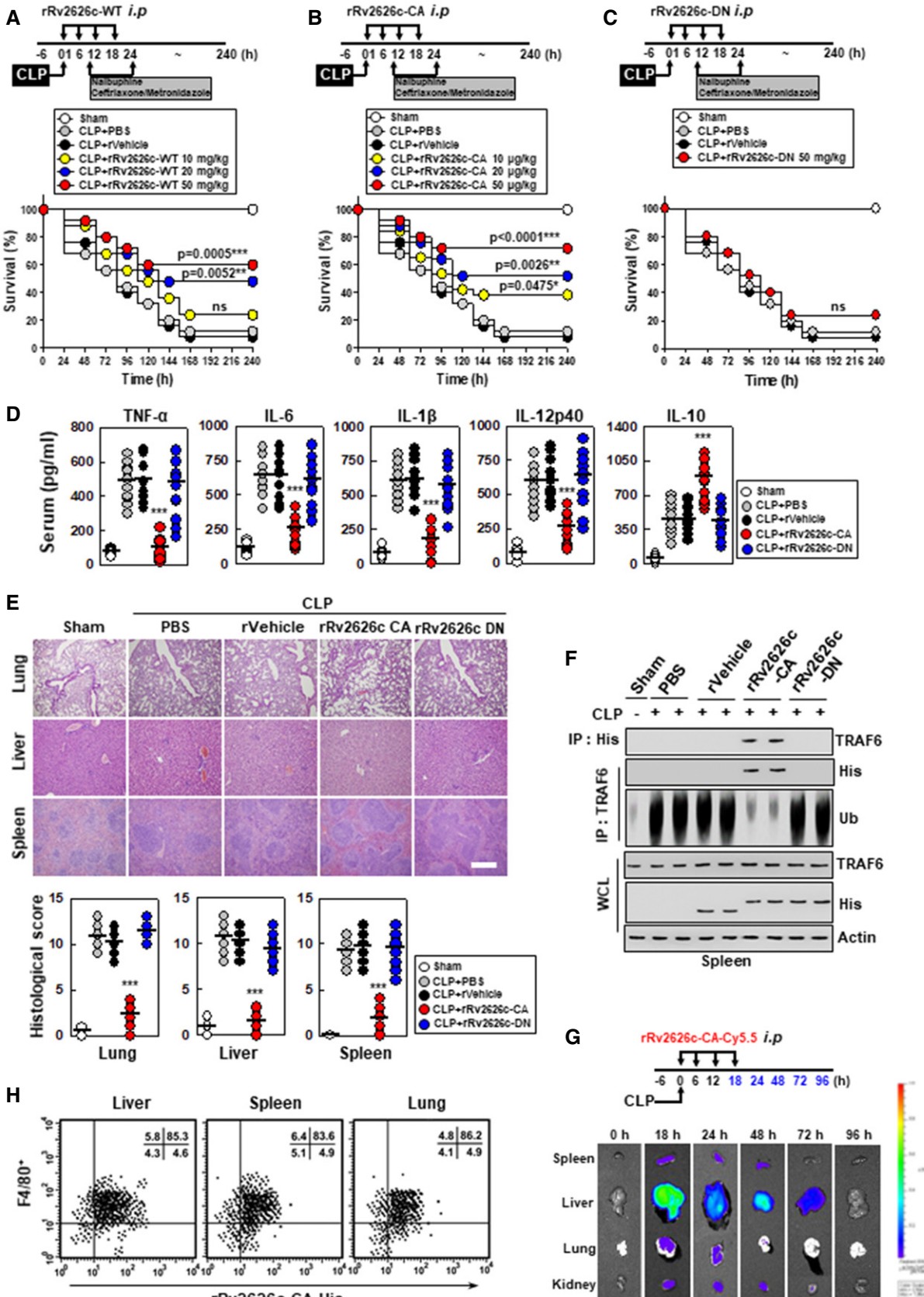

**Figure 6.**

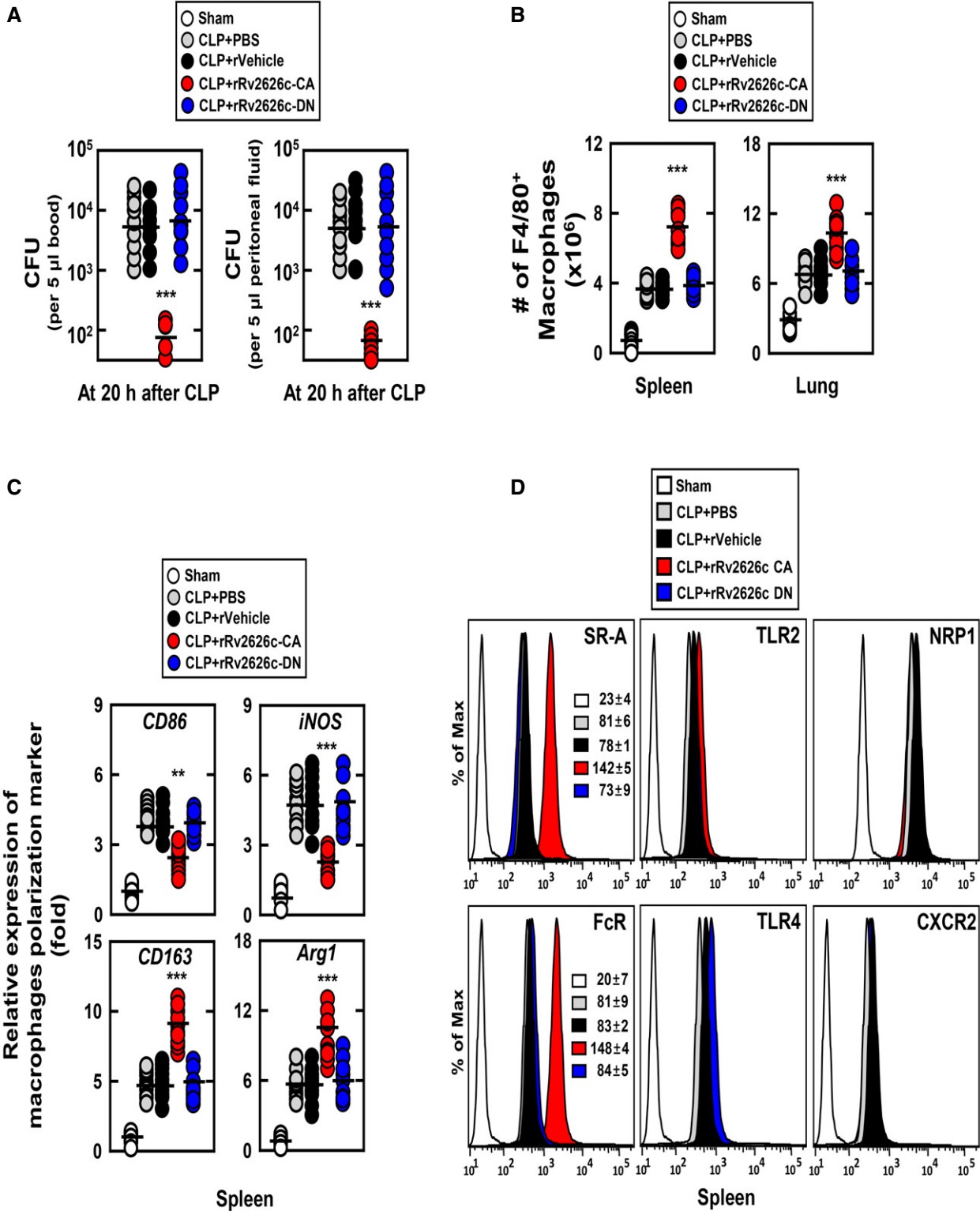

**Figure 7. rRv2626c-CA shows bactericidal effects.**

A  The bacterial burden was evaluated 20 h after treatment of CLP mice with rRv2626c-CA or its mutants (*n* = 10 mice per group).
B  The percentage of F4/80+ macrophages cells and rRv2626c-CA-His or its mutants were found in the spleen or lung using Fluorescence-activated cell sorting analysis in the background of CLP-induced sepsis.
C  CD86, iNOS, CD163, and Arg1 mRNA levels were determined by real-time PCR.
D  Expression of SR-A, FcR, TLR2, TLR4, NRP1, or CXCR2 was measured by flow cytometry analysis. The data are representative of five independent experiments with similar results (D). Statistical significance was determined by Student's *t*-test with Bonferroni adjustment (**P < 0.01; ***P < 0.001) compared with CLP+ rVehicle.

peptide (Fig 7 and Appendix Fig S3). Structural changes, such as folding and electron bonding, during protein expression appear to have resulted in the loss of the peptide's ability to control inflammation and bacteremia. There may also be functional differences between peptides and proteins (Jauset & Beaulieu, 2019). Nevertheless, its therapeutic effect in sepsis is due to the efficacy of the Rv2626c peptide ($_{123}$LPEHAIVQF$_{131}$), instead of the antibacterial activity of tuftsin; nevertheless, the potential effect of tuftsin on adaptive immunity remains unclear.

rRv2626c-CA is a potential therapeutic agent. However, therapeutic peptides and proteins have disadvantages, such as low bioavailability and metabolic liability. Moreover, they tend to have high molecular weight, low lipophilicity, and charged residues that hinder their absorption. The intraperitoneal delivery of these therapeutics overcomes the issue of absorption; however, other factors limit their. These factors include systemic proteases, rapid metabolism, opsonization, conformational changes, dissociation of protein subunits, noncovalent complex with blood products, and destruction of labile side groups (Bruno et al, 2013; Jauset & Beaulieu, 2019).

Oral delivery improves patient compliance; therefore, there is an increased interest in the development of systems that allow the oral delivery of therapeutic peptides and proteins (Aungst, 2012; Sam Maher, 2012). Further studies are warranted to determine the noninvasive delivery methods of rRv2626c-CA to overcome the obstacles preceding its application for the treatment of sepsis.

This study demonstrated the function of peptides in regulating innate immunity and provided a comprehensive description of the design, categorization, and application of peptide-based therapeutics for sepsis. Therapeutics peptides targeting macrophages have several advantages, including economic benefits, feasibility of clinical-grade manufacturing, and applicability to sepsis therapy. By determining the impact of peptide length and formulation on their immunogenicity, we showed that peptide-based agents can be used and lead to major breakthroughs. Moreover, challenges remain in large-scale synthesis, safe delivery, and efficient immunotherapy to improve next-generation peptide-based immunotherapy.

In conclusion, macrophage-targeted delivery and specificity for M2 immune cell type should be properly combined to maximize immunotherapeutic effects on sepsis, and future translational research and clinical trials are needed to promote peptide immunotherapeutics as next-generation sepsis immunotherapies.

## Materials and Methods

### Mice and cell culture

Wild-type C57BL/6 mice were purchased from Samtako Bio Korea (Gyeonggi-do, Korea). Primary bone marrow-derived macrophages (BMDMs) were isolated from C57BL/6 mice and cultured in DMEM for 3–5 days in the presence of M-CSF (R&D Systems, 416-ML), as described previously (Koh et al, 2017). BMDMs of TLR2$^{-/-}$, TLR4$^{-/-}$, MyD88$^{-/-}$, TRIF$^{-/-}$, IRAK1$^{-/-}$, TRAF6$^{-/-}$, and TBK1$^{-/-}$ in C57BL/6 mice were a generous gift from Dr. Chul-Ho Lee (Laboratory Animal Center, Korea Research Institute of Bioscience and Biotechnology, Daejeon, Korea). HEK293T (ATCC-11268; American Type Culture Collection) and RAW264.7 (ATCC TIB-71) in DMEM (Gibco)

containing 10% FBS (Gibco), sodium pyruvate, nonessential amino acids, penicillin G (100 IU/ml), and streptomycin (100 μg/ml). Human monocytic THP-1 (ATCC TIB-202) cells were grown in RPMI 1640/glutamax supplemented with 10% FBS and treated with 20 nM PMA (Sigma-Aldrich) for 24 h to induce their differentiation into macrophage-like cells, followed by washing three times with PBS. Transient transfections were performed using calcium phosphate (Clontech) in 293T, according to the manufacturer's instructions. RAW264.7 and THP-1 stable cell lines were generated by transfections were performed using Lipofectamine 3000 (Invitrogen) and then a standard selection protocol with 400–800 μg/ml of G418.

### Recombinant protein

To obtain recombinant MTB H37Rv strain Rv2626c (GenBank accession no. NP_217142.1) and CPP-Tuftsin-Rv2626c CBS2 (aa 123–131) protein, Rv2626c amino acid (1–143) and CPP seq. (GRKKRRQRR RPQ) macrophages targeting seq. (TKPR)-CBS2 (LPEHAIVQF) were cloned with an N-terminal 6xHis tag into the pRSFDuet-1 Vector (Novagen) and induced, harvested, and purified from Escherichia coli expression strain BL21(DE3)pLysS as described previously (Kim et al, 2020), in accordance with the standard protocols recommended by Novagen. rRv2626c was dialyzed with permeable cellulose membrane and tested for lipopolysaccharide contamination with a Limulus amebocyte lysate assay (BioWhittaker) and contained < 20 pg/ml at the concentrations of rRv2626c proteins used in the experiments described here.

### Reagents and antibodies

LPS (Escherichia coli O111:B4) and BLP (Pam$_2$CSK$_4$) were purchased from Invivogen. Specific Abs against phospho-(Ser473)-AKT (Dilution 1:2,000), phospho-(Thr202/Tyr204)-p42/44 (Dilution 1:2,000), phospho-(Thr180/Tyr182)-p38 (Dilution 1:2,000), phospho-(Thr183/ Tyr185)-SAPK/JNK (Dilution 1:2,000), phospho-(Ser32/36)-IκB-α (Dilution 1:2,000), K63-Poly-Ub (D7A11, dilution 1:1,000), and K48-Poly-Ub (D9D5, dilution 1:1,000) were purchased from Cell Signaling Technology (Danvers, MA, USA). Abs specific for IκB-α (C-21, dilution 1:1,000), TRAF6 (H-274, dilution 1:1,000), Lamin B1 (B-10, dilution 1:1,000), Tubulin (5F131, dilution 1:1,000), CD68 (KP1, dilution 1:250), F4/80 (BM8, dilution 1:250), CD3 (PC3/188A, dilution 1:250), CD19 (SJ25-C1, dilution 1:250), Ub (P4D1, dilution 1:1,000), His (His17, dilution 1:1,000), HA (12CA5, dilution 1:1,000), Flag (D-8, dilution 1:1,000), GST (B-14, dilution 1:1,000), Myc (9E10, dilution 1:1,000), and Actin (I-19, dilution 1:5,000) were purchased from Santa Cruz Biotechnology.

### Plasmid construction

HA-tagged ubiquitin (Ub), K48-linkage specific ubiquitin (K48-Ub), and K63-linkage specific ubiquitin (K63-Ub) plasmids were purchased from Addgene. The plasmid encoding full length of the TRAF6 and mutant plasmids were previously described (Yang et al, 2016). Plasmids encoding different regions of Rv2626c (1–143, 8–65, 73–131 in Table 1) were generated by PCR amplification from full-length Rv2626c cDNA and subcloning into a pEBG derivative encoding an N-terminal GST epitope tag between the BamHI and NotI sites. All constructs for transient

and stable expression in mammalian cells were derived from the pEBG-GST mammalian fusion vector and the pEF-IRES-Puro expression vector. All constructs were sequenced using an ABI PRISM 377 automatic DNA sequencer to verify 100% correspondence with the original sequence.

### Peptides

Tat-conjugated Rv2626c peptides were commercially synthesized and purified in acetate salt form to avoid abnormal responses in cell by Peptron (Korea). The amino acid sequences of the peptides in this study are described in Table 2. The endotoxin content as measured by the Limulus amebocyte lysate assay (BioWhittaker) and contained less than 3–5 pg/ml at the concentrations of the peptides used in experiments.

### GST pulldown, immunoblot, and immunoprecipitation analysis

GST pulldown, immunoprecipitation, and immunoblot assays were performed as previously described (Koh *et al*, 2017; Kim *et al*, 2018).

For GST pulldown, cells were harvested and lysed in NP-40 buffer supplemented with a complete protease inhibitor cocktail (Roche). After centrifugation, the supernatants were precleared with protein A/G beads at 4°C for 2 h. Precleared lysates were mixed with a 50% slurry of glutathione-conjugated Sepharose beads (Amersham Biosciences), and the binding reaction was incubated for 4 h at 4°C. Precipitates were washed extensively with lysis buffer. Proteins bound to glutathione beads were eluted with SDS loading buffer by boiling for 5 min.

For immunoprecipitation, cells were harvested and then lysed in NP-40 buffer supplemented with a complete protease inhibitor cocktail (Roche). After preclearing with protein A/G agarose beads for 1 h at 4°C, whole-cell lysates were used for immunoprecipitation with the indicated antibodies. Generally, 1–4 μg of commercial antibody was added to 1 ml of cell lysates and incubated at 4°C for 8–12 h. After the addition of proteins A/G agarose beads for 6 h, immunoprecipitates were extensively washed with lysis buffer and eluted with SDS loading buffer by boiling for 5 min.

For immunoblotting, polypeptides were resolved by SDS–polyacrylamide gel electrophoresis (PAGE) and transferred to a PVDF membrane (Bio-Rad). Immunodetection was achieved with specific antibodies. Antibody binding was visualized by chemiluminescence (ECL; Millipore) and detected by a Vilber chemiluminescence analyzer (Fusion SL 3; Vilber Lourmat).

### Enzyme-linked immunosorbent assay

Cell culture supernatants and mice sera were analyzed for cytokine content using the BD OptEIA ELISA set (BD Pharmingen) for the detection of TNF-α, IL-6, IL-1β, IL-12p40, and IL-10. All assays were performed as recommended by the manufacturer.

### CLP-induced sepsis and bacteria counts

Cecal ligation and puncture (CLP) were performed using 6-week-old C57BL/6 female mice (Samtako Bio, Gyeonggi-do, Korea), as described previously (Kim *et al*, 2016; Kim *et al*, 2018). For CLP, mice were anesthetized with pentothal sodium (50 mg/kg, *i.p.*), and

a small abdominal midline incision was made to expose the cecum. The cecum was then ligated below the ileocecal valve, punctured twice through both surfaces, using a 22-gauge needle, and the abdomen was closed. The survival rate was monitored daily for 10 days. The mice were resuscitated by intraperitoneal injection of PBS, analgesic (1.5 mg/kg nalbuphine; Sigma-Aldrich), and an antibiotic cocktail containing ceftriaxone (25 mg/kg; Sigma-Aldrich) and metronidazole (12.5 mg/kg; Sigma-Aldrich) in 100 μl PBS at 12 and 24 h after CLP onset (Jeger *et al*, 2016; Van Wyngene *et al*, 2020). For experiments aimed to isolate blood and organ samples, sham-operated mice of which the cecum was exposed but not ligated or punctured were used and are indicated as sham at the time of the surgery.

For bacteria count, blood was collected by cardiac puncture or peritoneal lavage fluids from mice at indicated time after CLP. After performing serial dilution of blood, 5 μl of each dilution was plated on blood agar plates. Bacteria were counted after incubation at 37°C for 24 h and calculated as counting colony-forming units per blood or whole peritoneal lavage.

All animals were maintained in a pathogen-free environment. All animal experimental procedures were reviewed and approved by the Institutional Animal Care and Use Committee of Hanyang University (protocol 2018-0086). CLP model that post-CLP analgesia, fluid support, and antibiotics is consistent with international guidelines, defined as the "Minimum Quality Threshold in Pre-Clinical Sepsis Studies" for the sepsis mouse model, to enhance translational relevance of the models (Mai *et al*, 2018; Zingarelli *et al*, 2019).

### Histology

For immunohistochemistry of tissue sections, mouse spleens, livers, and lungs were fixed in 10% formalin and embedded in paraffin. Paraffin sections (4 μm) were cut and stained with hematoxylin and eosin (H&E). Histopathologic score was established on the basis of the numbers and distribution of inflammatory cells and the severity of inflammation within the tissues (Kennedy *et al*, 2000; Buchweitz *et al*, 2007) in which a board-certified pathologist independently scored each organ section without prior knowledge of the treatment groups. A histological score ranging from 0 to 4 was ascribed to each specimen.

### *In vivo* imaging

rRv2626c-CA/Cy5.5 was prepared by adding streptavidin-conjugated Cy5.5 dye to rRv2626c-CA. rRv2626c-CA/Cy5.5 were administered into mouse via *i.p* in CLP mice. To study tissue biodistribution, mice were sacrificed at different time points post-administration and the major organs were excised and imaged using the IVIS Spectrum-CT *in vivo* imaging system (PerkinElmer, Inc.).

### Protein purification and mass spectrometry

To identify rRv2626c-binding proteins, THP-1 cells were treated with or without rRv2626c for 30 min, harvested, and lysed with NP-40 buffer (50 mM HEPES, pH 7.4, 150 mM NaCl, 1 mM EDTA, and 1% (*v/v*) NP40) supplemented with a complete protease inhibitor cocktail (Roche). Post-centrifuged supernatants

**The paper explained**

**Problem**
Mortality of Gram-negative bacterial sepsis caused by hyper-activation of TLR4 signaling pathway remains extremely high despite the vigorous investigation to develop new therapeutic interventions targeting both pathogens and host immune responses. Due to the lack of innovative insights and therapeutics, sepsis remains a high unmet medical need. In our opinion, this lack of treatment is largely due to the fact that inflammation and bacteria have been considered as the main driving and killing mechanism in sepsis. However, recent evidence suggests profound research into therapeutic approaches that target macrophages, a major site of sepsis.

**Results**
In our paper, we found mycobacteria Rv2626c (C-terminal 123–131-amino acid) interacts with the TRAF6 and inhibits E3 ubiquitin ligase activity, thereby suppressing TLR4 inflammatory signaling in macrophages. We generated recombinant proteins of macrophage-targeting peptide (Tuftsin)-conjugated peptide composed of nine amino acids 123–131 of Rv2626c, named rRv2626c-CA. We found that rRv2626c-CA had a significant therapeutic efficacy in mouse cecal ligation–puncture (CLP)-induced sepsis. C-terminal 123–131-amino acid Rv2626c motif promotes anti-inflammation, macrophage recruitment, phagocytosis, M2 macrophage polarization, and subsequent bacterial clearance.

**Impact**
These data show that mycobacteria Rv2626c, in addition to influencing inflammation, plays a crucial role in the control of polymicrobial during sepsis. Moreover, we have clearly demonstrated that multifunctional rRv2626c-CA activity during sepsis, causing macrophage environment alteration. We believe our findings may have a major impact on the sepsis research field and could pave the way to new therapeutic interventions in sepsis as our data using a novel rRv2626c-CA are promising.

were precleared with protein A/G beads at 40°C for 2 h. Precleared lysates were mixed with αHis antibody conjugated with agarose beads for 4 h at 4°C. Precipitates were washed extensively with lysis buffer. Proteins bound to beads were eluted and separated on a Nupage 4–12% Bis-Tris gradient gel (Invitrogen). After silver staining (Invitrogen), specific protein bands were excised and analyzed by ion-trap mass spectrometry at the Korea Basic Science Institute Mass Spectrometry facility, and amino acid sequences were determined by tandem mass spectrometry and database searches.

**Quantitative real-time polymerase chain reaction**

Total RNA was extracted from cells using an RNeasy RNA extraction Mini-Kit (Qiagen). cDNA was synthesized using an Enzynomix kit (Enzynomix), and quantitative PCR was performed using gene-specific primer sets (Bioneer) and SYBR Green PCR Master Mix (Roche). Real-time PCR was performed using a QuantStudio™ 3 (ABI), according to the manufacturer's instructions. Data were normalized to the expression of β-actin. Relative expression was calculated using the delta–delta Ct method. The sequences of the primers were as follows: mCD86 (Forward: gcacgtctaagcaaggtcac;

Reverse: catatgccacacaccatccg), miNOS (Forward: ccccgctactactccatcag; Reverse: ccactgacacttcgcacaaa), mCD163 (Forward: tgtgaccatgctgaggatgt; Reverse: ctcgaccaatggcactgatg), mArg1 (Forward: ctgagctttgatgtcgacgg; Reverse: tcctctgctgtcttcccaag), mβ-Actin (Forward: aagtgtgacgttgacatc; Reverse: gatccacatctgctggaagg).

**Confocal fluorescence microscopy**

Immunofluorescence analysis was performed as described previously (Koh *et al*, 2017). The cells were fixed on coverslips with 4% (*w/v*) paraformaldehyde in PBS and then permeabilized for 10 min using 0.25% (*v/v*) Triton X-100 in PBS at 25°C. TRAF6 or His was detected using a 1/100 dilution of the primary Ab for 1 h at 25°C. After washing, the appropriate fluorescently labeled secondary Abs were incubated for 1 h at 25°C. Slides were examined using laser-scanning confocal microscopy (model LSM 800; Zeiss).

**Cellular fractionation**

Cytosol and mitochondria were isolated from cells using a Mitochondria Fractionation Kit (Active Motif, 40015) or as described previously (Kim *et al*, 2020). Subcellular fractionated proteins were lysed in buffer containing 2% SDS and boiled with 2× reducing sample buffer for SDS–PAGE.

**MTT assay**

Cell viability relative to nontreated group was measured by MTT assay, as described previously (Kim *et al*, 2020). After incubating for the indicated time points, 5 mg/ml of MTT (3-(4,5-dimethylthiazol-2-yl)-2,5-diphenyltetrazolium bromide) solution was added in the place of media, and cells were incubated for further 4 h. Then, all the media was removed and the same volume of dimethyl sulfoxide (DMSO) solution was added for 15 min to dissolve the formazan. Using UV/VIS spectrophotometer, each well of the plate was measured at 540 nm to measure relative cell viability.

**Flow cytometry**

Flow cytometry data were acquired on a FACSCanto (BD Biosciences, San Diego, CA) and analyzed with FlowJo software (Tree Star, Ashland, OR). To determine expression of cell surface proteins, mAb were incubated at 4°C for 20–30 min and cells were fixed using Cytofix/Cytoperm Solution (BD Biosciences) and in some instances followed by mAb incubation to detect intracellular proteins. The following mAb clones were used: NK1.1 (PK136, eBioscience), LY6G (1A8-Ly6g eBioscience), SR-A (PSL204, eBioscience), FcR (MAR-1, eBioscience), TLR2 (6C2, eBioscience), TLR4 (HTA125, eBioscience), NRP1 (3DS304M, eBioscience), CXCR2 (eBio5E8-C7-F10 (5E8-C7-F10), eBioscience).

**Statistical analysis**

Randomization, blinding, or replication was not applicable in this study. Continuous variables were described as means and standard errors, or medians and interquartile range values. Categorical variables were expressed as counts and percentages. All data were analyzed using Student's *t*-test with Bonferroni adjustment or

ANOVA for multiple comparisons and are presented as mean ± SD. Statistical analyses were conducted using the SPSS (version 12.0) statistical software program (SPSS, Chicago, IL, USA). Differences were considered significant at $P < 0.05$. For survival, data were graphed and analyzed by the product limit method of Kaplan and Meier, using the log-rank (Mantel–Cox) test for comparisons using GraphPad Prism (version 5.0, La Jolla, CA, USA). Respective statistical tests used are stated in the main text and figure legends. All $P$-values for main Figures and appendix figures can be found in Appendix Table S1.

**Expanded View** for this article is available online.

## Data availability

This study includes no data deposited in external repositories.

**Expanded View** for this article is available online.

## Acknowledgements

This work was supported by the NRF grant funded by the Korea government (MSIP) (2016R1D1A1A02937312 and 2019R1I1A2A01064237); a grant from the KHIDI, funded by the Ministry of Health & Welfare, Republic of Korea (HI16C1653). We would like to thank all members of the Infection Biology Lab for critical reading and discussion of the manuscript.

## Author contributions

SYK, DK, SK, DL, SJM, EC, and WS performed molecular and animal experiments and also analyzed data. KJ analyzed the histological data of tissue sections. CSY designed and conceptualized the research, supervised the experimental work, analyzed data, and wrote the manuscript.

## Conflict of interest

The authors declare that they have no conflict of interest.

## For more information

The URLs for data presented in this article are as follows:

(i)      1,000 genomes, http://www.1000genomes.org
(ii)     dbSNP, http://www.ncbi.nlm.nih.gov/projects/SNP
(iii)    Ensembl, http://www.ensembl.org
(iv)    European Genome-Phenome Archive (EGA), https://www.ebi.ac.uk/ega
(v)     Primer 3, http://frodo.wi.mit.edu/primer3
(vi)    PRoteomics IDEntifications database (PRIDE), https://www.ebi.ac.uk/pride
(vii)   RefSeq, https://www.ncbi.nlm.nih.gov/refseq
(viii)  UCSC Genome Bioinformatics, http://genome.ucsc.edu
(ix)    UniProt, http://www.uniprot.org

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
