## [Review Process File · EMBO Molecular Medicine]

Mycobacterium tuberculosis Rv2626c-derived peptide as a therapeutic agent for sepsis

Sun Young Kim, Donggyu Kim, Sojin Kim, Daeun Lee, Seok-Jun Mun, Eun-I Cho, Wooic Son, Kiseok Jang, Chul-Su Yang

DOI: [10.15252/emmm.202012497](https://doi.org/10.15252/emmm.202012497)

Corresponding author: Chul-Su Yang (chulsuyang@hanyang.ac.kr)

Review Timeline:

Submission Date:	10th Apr 20
Editorial Decision:	20th Apr 20
Revision Received:	28th Apr 20
Editorial Decision:	16th Jun 20
Revision Received:	1st Sep 20
Editorial Decision:	13th Oct 20
Revision Received:	22nd Oct 20
Accepted:	27th Oct 20

Editor: Zeljko Durdevic

Transaction Report:

20th Apr 2020

Dear Prof. Yang,

Thank you for submitting your manuscript to EMBO Molecular Medicine. I have now carefully read your article and discussed it with the other members of our editorial team. I am happy to inform you that we find the manuscript interesting and novel and would therefore like to proceed with peer review.

However, before we invite referees to assess your study, we would like to suggest that you revise your article for grammar and syntax (i.e. by an English native speaker), in order to adequately convey the important messages of your manuscript. As an example, the reader might understand that rRv2626c inhibits ubiquitination of TRAF6, while rRv2626c actually blocks TRAF6 ubiquitin ligase activity. Furthermore, your manuscript would also benefit from a minor reorganization of the figures to improve coherence between the text and the presented data. We also encourage you to check the color labeling of the figures for potential inconsistencies (e.g. figure 6D, IL-10 graph or 7A). We feel that these minor revisions would help the referees to focus on the important findings of your work.

I hope you will find our suggestion reasonable and I look forward to receiving your revised manuscript.

The authors performed the requested changes.

16th Jun 2020

Dear Prof. Yang,

Thank you for the submission of your manuscript to EMBO Molecular Medicine. We have now heard back from the three referees who agreed to evaluate your manuscript. As you will see from the reports below, the referees acknowledge the interest of the study. However, they raise some concerns that should be addressed in a major revision of the present manuscript. Particular attention should be given to the validation of the findings in a clinically more relevant, less lethal mouse model of sepsis. Addressing the reviewers' concerns in full will be necessary for further considering the manuscript in our journal.

Acceptance of the manuscript will entail a second round of review. Please note that EMBO Molecular Medicine encourages a single round of revision only and therefore, acceptance or rejection of the manuscript will depend on the completeness of your responses included in the next, final version of the manuscript. For this reason, and to save you from any frustrations in the end, I would strongly advise against returning an incomplete revision.

We realize that the current situation is exceptional on the account of the COVID-19/SARS-CoV-2 pandemic. Therefore, please let us know if you need more than three months to revise the manuscript.

I look forward to receiving your revised manuscript.

***** Reviewer's comments *****

Referee #1 (Comments on Novelty/Model System for Author):

The lethal CLP model that does not use fluid resuscitation, analgesia and antibiotics is not appropriate for human sepsis. The model used here has 100% lethality in 48 hours, and human sepsis has mortality of less than 30% an death usually occurs after 14 days. The current model has been rejected by MQTiPSS, a consortium that has set minimal standards for CLP/sepsis mouse

studies.

Referee #1 (Remarks for Author):

This is an extremely interesting exploration of a novel peptide derived from *M. tuberculosis* that has potent anti-inflammatory properties. Using a series of detailed in vitro studies with BMDM obtained from various KO mice, the authors convincingly demonstrate the signaling pathways by which this peptide acts.

The in vitro studies are clearly the strength of the studies and build on the extensive experience of the investigators. The authors move to in vivo studies in mice and demonstrate that their new construct has approximately 3-logs greater potency than previous constructs.

Unfortunately, the manuscript falls apart when the authors move to the in vivo studies where there are significant concerns about the model they have selected. The authors used a model of CLP that is uniformly lethal, and lethality is rapid, within 48 hours. The model does not use fluid resuscitation, antibiotics and analgesics. Importantly, this is a model not representative of human sepsis, but more closely reflective of endotoxic or bacteremic shock. The human sepsis world is not recapitulated by these high lethality models, and they have lost favor in the field. Human sepsis, even septic shock, has an in hospital mortality of at most these days 30%.

This is not sepsis, but is endotoxic or bacteremic shock. These studies, to have any clinical relevance, need to be performed in a CLP model that includes fluids, analgesia and antibiotics and has a 7-14 days mortality of at most 50%. This will mean reducing the size of the enterotomy.

The authors should understand that the murine model of CLP has attracted a great deal of controversy, and is generally no longer accepted in as a sole model of sepsis. If the authors are interested in further developing the drug for clinical use, they need to consider testing this drug in less lethal and multiple sepsis models, including CLP, pneumonia, gram positive infections.

<https://www.nigms.nih.gov/News/reports/Documents/nagmsc-working-group-on-sepsis-final-report.pdf>

Finally, the authors need to be aware that every sepsis trial that has used an anti-inflammatory approach has failed in human randomized clinical trials. Not a single anti-inflammatory agent has been approved for human sepsis. Whether this is because previous drugs were not as effective as this molecule, or that human sepsis is not amenable to anti-inflammatory treatment is unknown.

Specific Concerns:

1. The absence of post CLP analgesia, fluid support and antibiotics is not consistent with international guidelines for the sepsis mouse model (Shock. 2019 Jan;51(1):10-22.). Although the protocols may have been approved by the local ethics committee, that in itself is not sufficient for widespread acceptance. The authors need to state whether they adhere to MQTiPSS.
2. Were the purified rRv2626c (16 and 36 kDa) tested for LPS contamination with LAL?
3. Could the negative inflammatory responses of rRv2626c simply be due to tolerance and cross-tolerance since pretreatment activated MyD88 signaling pathways.

Referee #2 (Remarks for Author):

This paper is of interest.

But the novelty is compromised by previous work, e.g. the 2018 J. Immunol. paper 'Mycobacterium tuberculosis PPE18 Protein Reduces Inflammation and Increases Survival in Animal Model of Sepsis'

The current paper provides more mechanistic insight in the protective effects of this Mycobacterium protein in sepsis and the data are well presented and of interest.

Since the authors couple the mycobacterium peptide (Rv2626c) to tuftsin to target macrophages, but since tuftsin has immunoregulatory functions, it is not Always clear what we are looking at.

1. The experiments appear to be well-performed, however some control groups are lacking:
 - Figure 5D: solvent (medium?) and rvehicle (=tuftsin) as control groups are lacking here. Does rvehicle alone already have some anti-inflammatory effects? Moreover, the reasoning as to why combining Rv2626c to tuftsin is not clear to me. Is it simply because tuftsin targets the macrophages and makes the peptide thus more cell-specific? Or do the authors combine tuftsin to RV2626c as tuftsin also have immunomodulatory effects? This needs more explanation in the results and discussion. In line with this, how do the authors explain the improved potency of the conjugated peptide?
 - Also for the in vivo studies, control groups are lacking. For the in vivo experiments, rvehicle (=tuftsin) is used as a control. However this rvehicle (=tuftsin) may have antimicrobial effects, so an extra control group is lacking, namely mice injected with vehicle (PBS?) alone to see the effect of tuftsin on survival, cytokine levels, organ damage and phagocytosis.
 - To detect the effects of the CLP surgery on the above described parameters, a sham group has to be taken along in all the in vivo experiments
2. Is protection with rRv2626CA against CLP-induced lethality lost in TLR2KO, MyD88KO, TRAFKO, and/or IRAK KO mice? I understand that testing the peptide in all these lines requires a lot of mice and time. However, minimal confirmation of the in vitro experiments would be informative. For example test the peptide in TRAF6 KO mice.
3. In figure 6D and 7A, the authors show the data as individual data points, however in figure 6E and 7B-C, the data are grouped. I prefer to show all the data points individually.
4. Page 17: "increased H&E staining" needs rephrasing
5. The discussion part is comparatively poor; the authors mainly repeat the overall findings of the study. This needs to be revised. What are the advantages of the current described peptide with for example the MTB PPE described by Ahmed et al? Is it the addition of tuftsin that distinguishes this peptide based on its antibacterial capacity? I also do not understand the sudden mentioning of chemotherapy here.

Referee #3 (Remarks for Author):

Kim et al. determined the effect of Rv2626c, a peptide derived from Mycobacterium tuberculosis. Considering, that this peptide has been shown to elicit a strong serum antibody response in patients with tuberculosis, the authors focussed on analysing the signalling cascade induced by a HIS-tagged version of this peptide. Kim et al. showed that Rv2626c binds to the RING domain of

TRAF6, inhibiting its lysine 63-linked polyubiquitination, which consequently blocked TLR4 signalling in macrophages. However, RV2626c cellular uptake required TLR2. Additionally, the authors observed that the C-terminal 123-131 amino acids motif of Rv2626c led to macrophage recruitment, phagocytosis, M2 polarization, and bacterial clearance. In vivo, the authors tested the Rv2626c peptide as a constitutive active (CA) and dominant negative (DN) form. With this setting, the authors observed an increased survival of mice following polymicrobial sepsis initiation.

Although the author's study is well presented and well designed, some concerns should be addressed:

Materials and Methods:

CLP-induced sepsis and Bacteria counts

"For bacteria count, Blood and peritoneal lavage fluids were collected from mice by cardiac puncture..."

"Blood" should be "blood". Peritoneal lavage cannot be taken by cardiac puncture. Please correct.

Miscellaneous Procedures

All words (apart from "Details" and "Supporting Information") should be lowercase.

Figure 1A: Is the dimerization due the HIS-tag? Is the effect observed caused by the monomer or the dimer?

Figure 1B: To justify the term "BMDMs" more than one cell should be shown.

Figure 3: Have the THP1 cells been differentiated to macrophages or have these cells been used directly? If not, this should be specified in the manuscript.

Figure 6G: The times following CLP in the schematic picture in the upper part of the Figure should be adapted to the times in the lower part of the picture.

***** Reviewer's comments *****

Referee #1 (Comments on Novelty/Model System for Author):

The lethal CLP model that does not use fluid resuscitation, analgesia and antibiotics is not appropriate for human sepsis. The model used here has 100% lethality in 48 hours, and human sepsis has mortality of less than 30% and death usually occurs after 14 days. The current model has been rejected by MQTiPSS, a consortium that has set minimal standards for CLP/sepsis mouse studies.

Referee #1 (Remarks for Author):

This is an extremely interesting exploration of a novel peptide derived from *M. tuberculosis* that has potent anti-inflammatory properties. Using a series of detailed in vitro studies with BMDM obtained from various KO mice, the authors convincingly demonstrate the signaling pathways by which this peptide acts.

The in vitro studies are clearly the strength of the studies and build on the extensive experience of the investigators. The authors move to in vivo studies in mice and demonstrate that their new construct has approximately 3-logs greater potency than previous constructs.

Unfortunately, the manuscript falls apart when the authors move to the in vivo studies where there are significant concerns about the model they have selected. The authors used a model of CLP that is uniformly lethal, and lethality is rapid, within 48 hours. The model does not use fluid resuscitation, antibiotics and analgesics. Importantly, this is a model not representative of human sepsis, but more closely reflective of endotoxic or bacteremic shock. The human sepsis world is not recapitulated by these high lethality models, and they have lost favor in the field. Human sepsis, even septic shock, has an in hospital mortality of at most these days 30%.

This is not sepsis, but is endotoxic or bacteremic shock. These studies, to have any clinical relevance, need to be performed in a CLP model that includes fluids, analgesia and antibiotics and has a 7-14 days mortality of at most 50%. This will mean reducing the size of the enterotomy.

The authors should understand that the murine model of CLP has attracted a great deal of controversy, and is generally no longer accepted in as a sole model of sepsis. If the authors are interested in further developing the drug for clinical use, they need to consider testing this drug in less lethal and multiple sepsis models, including CLP, pneumonia, gram positive infections.

<https://www.nigms.nih.gov/News/reports/Documents/nagmsc-working-group-on-sepsis-final-report.pdf>

Finally, the authors need to be aware that every sepsis trial that has used an anti-inflammatory approach has failed in human randomized clinical trials. Not a single anti-inflammatory agent has been approved for human sepsis. Whether this is because previous drugs were not as effective as this molecule, or that human sepsis is not amenable to anti-inflammatory treatment is unknown.

Specific Concerns:

1. The absence of post CLP analgesia, fluid support and antibiotics is not consistent with international guidelines for the sepsis mouse model (Shock. 2019 Jan;51(1):10-22.). Although the protocols may have been approved by the local ethics committee, that in itself is not sufficient for widespread acceptance. The authors need to state whether they adhere to MQTiPSS.

⇒ Thanks for your kind and valuable comments. We did the experiment again using a protocol that followed the international guidelines defined as the "Minimum Quality Threshold in Pre-Clinical Sepsis Studies" for the sepsis mouse model, to enhance translational relevance of the models, as suggested by the editor and reviewer in revised version.

In Fig. 6 and 7, We added various controls and sham conditions during the revision experiment. The results are clear and more reproducible than the original version results.

We added in Materials and Methods of revised manuscript, as below.

CLP-induced sepsis and bacteria counts

Cecal ligation and puncture (CLP) were performed using 6-week-old C57BL/6 female mice (Samtako Bio, Gyeonggi-do, Korea), as described previously (Kim et al., 2018, Kim et al., 2016). For CLP, mice were anesthetized with pentothal sodium (50 mg/kg, *i.p.*), and a small abdominal midline incision was made to expose the cecum. The cecum was then ligated below the ileocecal valve, punctured twice through both surfaces, using a 22-gauge needle, and the abdomen was closed. The survival rate was monitored daily for 10 days. The mice were resuscitated by intraperitoneal injection of PBS, analgesic (1.5 mg/kg nalbuphine; Sigma-Aldrich), and an antibiotic cocktail containing ceftriaxone (25 mg/kg; Sigma-Aldrich) and metronidazole (12.5 mg/kg; Sigma-Aldrich) in 100 μ l PBS at 12 h and 24 h after CLP onset (Jeger, Haufler et al., 2016, Van Wyngene, Vanderhaeghen et al., 2020). For experiments aimed to isolate blood and organ samples, sham-operated mice of which the cecum was exposed but not ligated or punctured were used and are indicated as sham at the time of the surgery.

For bacteria count, blood was collected by cardiac puncture or peritoneal lavage fluids from mice at indicated time after CLP. After performing serial dilution of blood, 5 μ l of each dilution was plated on blood agar plates. Bacteria were counted after incubation at 37 °C for 24 h and calculated as counting colony-forming units per blood or whole peritoneal lavage.

All animals were maintained in a pathogen-free environment. All animal experimental procedures were reviewed and approved by the Institutional Animal Care and Use Committee of Hanyang University (protocol 2018-0086). CLP model that post-CLP analgesia, fluid support and antibiotics is consistent with international guidelines, defined as the "Minimum Quality Threshold in Pre-Clinical Sepsis Studies" for the sepsis mouse model, to enhance translational relevance of the models (Mai, Sharma et al., 2018, Zingarelli, Coopersmith et al., 2019).

2. Were the purified rRv2626c (16 and 36 kDa) tested for LPS contamination with LAL?

⇒ Thanks for your kind comments. We did LAL assay and LPS contamination is a negligible amount. We added in Materials and Methods of revised manuscript, as below.

rRv2626c was dialyzed with permeable cellulose membrane and tested for lipopolysaccharide contamination with a *Limulus* amoebocyte lysate assay (BioWhittaker) and contained < 20 pg/ml at the concentrations of rRv2626c proteins used in the experiments described here.

Furthermore, rRv2626c-induced proinflammatory cytokines secretion was not significantly decreased in TLR4^{-/-} BMDMs compared with WT (Fig. 1E).

3. Could the negative inflammatory responses of rRv2626c simply be due to tolerance and cross-tolerance since pretreatment activated MyD88 signaling pathways.

⇒ Thanks for your kind and excellent comments. rRv2626c is internalized into the macrophages through TLR2 (Fig. 1F). Mass spectrometry revealed that bind to rRv2626c including TLR2 and TRAF6 (Fig. 3A), not MyD88. rRv2626c-induced the production of proinflammatory cytokines (TNF- α , IL-6, and IL-12p40) and anti-inflammatory cytokines (IL-10) through the TLR2/MyD88/TRAF6/IRAK1 pathway in macrophages (Fig. 1E) but it is a small concentration compared to the inflammatory response by LPS.

In TLR4/LPS-induced inflammatory signaling pathways, in the presence of rRv2626c, rRv2626c enters into the cytosol via TLR2-MyD88 dependent pathway. Cytosolic rRv2626c binds to N-terminal region of TRAF6 through CBS2 domain and inhibits K63-linked polyubiquitination (E3 ubiquitin ligase activity). Consequentially, rRv2626c negatively regulates LPS and sepsis-mediated TLR4 signaling pathway.

Referee #2 (Remarks for Author):

This paper is of interest.

But the novelty is compromised by previous work, e.g. the 2018 J. Immunol. paper 'Mycobacterium tuberculosis PPE18 Protein Reduces Inflammation and Increases Survival in Animal Model of Sepsis'

The current paper provides more mechanistic insight in the protective effects of this Mycobacterium protein in sepsis and the data are well presented and of interest.

Since the authors couple the mycobacterium peptide (Rv2626c) to tuftsin to target macrophages, but since tuftsin has immunoregulatory functions, it is not Always clear what we are looking at.

1. The experiments appear to be well-performed, however some control groups are lacking:

- Figure 5D: solvent (medium?) and rvehicle (=tuftsin) as control groups are lacking here. Does rvehicle alone already have some anti-inflammatory effects? Moreover, the reasoning as to why combining Rv2626c to tuftsin is not clear to me. Is it simply because tuftsin targets the macrophages and makes the peptide thus more cell-specific? Or do the authors combine tuftsin to RV2626c as tuftsin also have immunomodulatory effects? This needs more explanation in the results and discussion. In line with this, how do the authors explain the improved potency of the conjugated peptide?

⇒ Thanks for your kind and valuable comments. We added solvent (PBS) in Fig 5D and rVehicle (tuftsin) in Fig S3A of revised manuscript. Solvent (PBS) and rVehicle (tuftsin) had no effect on inflammatory responses in Fig 5D and Fig S3A *in vitro* and on mouse survival rate and inflammatory responses, etc in Fig 6 and Fig 7 *in vivo*.

In this study, Tuftsin targets the macrophages and makes the proteins thus more cell-specific. We used recombinant Tuftsin and recombinant Tuftsin-conjugated Rv2626c protein, unlike previous studies that used Tuftsin peptides, to improve the stability and targeting function of macrophages of the peptide (Fig S4). It seems that the inflammatory control effect of the peptide is lost due to structural changes such as folding and electrovalent bonding during the protein expression process. Functional distinctions may also be made between peptides and proteins.

We have changed these in revised manuscript as below.

In this study, we generated rTuftsin (rVehicle) and rTuftsin-conjugated Rv2626c protein, thus more cell-specific, unlike previous studies that used Tuftsin peptides, to improve the stability and targeting function of macrophages of the peptide (Fig S4). The rTuftsin lost the previously reported function of controlling inflammation and antibacterial activity of Tuftsin peptide (Fig. 7 and Appendix Fig. S3). Structural changes such as folding and electron bonding during protein expression appear to have lost the peptide's ability to control inflammation and antibacteria. There may also be functional differences between peptides and proteins (Jauset & Beaulieu, 2019). Nevertheless, its therapeutic effect in sepsis is due to the efficacy of the Rv2626c peptide ($_{123}$ LPEHAIVQF $_{131}$), instead of the antibacterial activity of tuftsin, but the potential effect of tuftsin on adaptive immunity remains unclear.

- Also for the *in vivo* studies, control groups are lacking. For the *in vivo* experiments, rvehicle (=tuftsin) is used as a control. However this rvehicle (=tuftsin) may have antimicrobial effects, so an extra control group is lacking, namely mice injected with vehicle (PBS?) alone to see the effect of tuftsin on survival, cytokine levels, organ damage and phagocytosis.

⇒ Thanks for your kind and valuable comments. We added solvent (PBS) and rVehicle (tuftsin) group in Fig 6 and 7 of revised manuscript. Solvent (PBS) and rVehicle (tuftsin) had no effect on mouse survival, cytokine levels, organ damage and phagocytosis in Fig 6 and Fig 7 *in vivo*.

- To detect the effects of the CLP surgery on the above described parameters, a sham group has to be taken along in all the *in vivo* experiments

⇒ Thanks for your kind and valuable comments. We added sham group in Fig 6 and Fig 7 of revised manuscript. In animal experiments, the sham group had no effect on mouse survival rate and inflammatory responses, etc.

2. Is protection with rRv2626CA against CLP-induced lethality lost in TLR2KO, MyD88KO, TRAFKO, and/or IRAK KO mice? I understand that testing the peptide in all these lines requires a lot of mice and time. However, minimal confirmation of the *in vitro* experiments would be informative. For example test the peptide in TRAF6 KO mice.

⇒ Thanks for your kind and valuable comments. We did ELISA of TNF- α and IL-6 in LPS stimulated in WT, TLR2^{-/-}, or TRAF6^{-/-} BMDMs in presence of rRv2626c-CA (Fig S3B). rRv2626c-CA did not inhibit the TNF- α and IL-6 production in TLR2^{-/-} and TRAF6^{-/-} BMDMs because it can be interacted with between Rv2626c and TLR2 and TRAF6.

3. In figure 6D and 7A, the authors show the data as individual data points, however in figure 6E and 7B-C, the data are grouped. I prefer to show all the data points individually.

⇒ Thanks for your kind and valuable comments. We have changed these in the Fig 6E and Fig 7B-C of revised manuscript as reviewer suggested.

4. Page 17: "increased H&E staining" needs rephrasing

⇒ We are sorry for our mistakes. We have corrected these in the Results of revised manuscript.

5. The discussion part is comparatively poor; the authors mainly repeat the overall findings of the study. This needs to be revised. What are the advantages of the current described peptide with for example the MTB PPE described by Ahmed et al? Is it the addition of tuftsin that distinguishes this peptide based on its antibacterial capacity? I also do not understand the sudden mentioning of chemotherapy here.

⇒ Thanks for your kind and valuable comments. We have changed these in revised manuscript as below.

A previous report suggested that MTB PPE18/Rv1196 protein also reduces inflammation and increases survival in an animal model of sepsis (Ahmed, Dolasia et al., 2018). Similar to the rRv2626c, PPE18 also binds to TLR2. However, PPE18 selectively downregulates proinflammatory immune responses which is phosphorylation of suppressor of cytokine signaling 3, which then physically interacts with the I κ B α -NF- κ B/rel complex, thus preventing phosphorylation and degradation of I κ B α and nuclear translocation of p50 and p65 NF- κ B and c-rel transcription factors. As a consequence of this, there is downregulation of transcription of NF- κ B-regulated genes such as IL-12 and TNF- α (Nair, Pandey et al., 2011). Furthermore, PPE18 increases secretion of an anti-inflammatory cytokine (IL-10) via activation of p38 MAPK and M2 macrophages in macrophages (Ahmed et al., 2018, Nair, Ramaswamy et al., 2009). Unlike rPPE18 (which is composed of 391 amino acids), rRv2626c (which is composed of 9 amino acids) has advantages in mechanismly rational design, intracellular delivery of proteins and stability in blood, and can reduce unexpected off-target side effects in animal experiments. Comparing the concentrations used in animal experiments, rRv2626c has considerably improved potency, with a 500-fold (*in vivo*) lower than that of rPPE18 (5 mg/kg). Furthermore, Rv2626c may function like antibiotics by improving bacterial clearance via M2 macrophage polarization and recruitment as well as enhanced phagocytosis in sepsis. Thus, the ability of the C-terminal 123–131-amino acid Rv2626c motif to dampen harmful factors and promote protective response in sepsis makes rRv2626c-CA a promising therapeutic agent for treating sepsis.

There is an increase in research focused on cell-targeting peptides. Tuftsin provides a strategy for delivering therapeutic agents specifically to macrophages. The primary function of tuftsin or tuftsin-like peptides is to enhance phagocyte respiratory burst, migration/chemotaxis ability, and antigen presentation in cells of monocytic origin including macrophages, neutrophils, microglia, and Kupffer cells, thereby increasing antimicrobial and antitumor activities (Gao, Yu et al., 2016, Liu et al., 2012, Wu et al., 2012). In animal models, tuftsin or tuftsin-like peptides were found to improve innate immunity and survival (Gao et al., 2016, Wardowska, Dzierzbicka et al., 2009, Wu et al., 2012).

In this study, we generated rTuftsin (rVehicle) and rTuftsin-conjugated Rv2626c protein, thus more cell-specific, unlike previous studies that used Tuftsin peptides, to improve the stability and targeting function of macrophages of the peptide (Fig S4). The rTuftsin lost the previously reported function of controlling inflammation and antibacterial activity of Tuftsin peptide (Fig. 7 and Appendix

Fig. S3). Structural changes such as folding and electron bonding during protein expression appear to have lost the peptide's ability to control inflammation and antibacteria. There may also be functional differences between peptides and proteins (Jauset & Beaulieu, 2019). Nevertheless, its therapeutic effect in sepsis is due to the efficacy of the Rv2626c peptide ($_{123}\text{LPEHAIVQF}_{131}$), instead of the antibacterial activity of tuftsin, but the potential effect of tuftsin on adaptive immunity remains unclear.

Referee #3 (Remarks for Author):

Kim et al. determined the effect of Rv2626c, a peptide derived from *Mycobacterium tuberculosis*. Considering, that this peptide has been shown to elicit a strong serum antibody response in patients with tuberculosis, the authors focussed on analysing the signalling cascade induced by a HIS-tagged version of this peptide. Kim et al. showed that Rv2626c binds to the RING domain of TRAF6, inhibiting its lysine 63-linked polyubiquitination, which consequently blocked TLR4 signalling in macrophages. However, Rv2626c cellular uptake required TLR2. Additionally, the authors observed that the C-terminal 123-131 amino acids motif of Rv2626c led to macrophage recruitment, phagocytosis, M2 polarization, and bacterial clearance. In vivo, the authors tested the Rv2626c peptide as a constitutive active (CA) and dominant negative (DN) form. With this setting, the authors observed an increased survival of mice following polymicrobial sepsis initiation.

Although the author's study is well presented and well designed, some concerns should be addressed:

Materials and Methods:

CLP-induced sepsis and Bacteria counts

"For bacteria count, Blood and peritoneal lavage fluids were collected from mice by cardiac puncture..."

"Blood" should be "blood". Peritoneal lavage cannot be taken by cardiac puncture. Please correct.

⇒ We are sorry for our mistakes. We have corrected these in the Materials and Methods of revised manuscript.

Miscellaneous Procedures

All words (apart from "Details" and "Supporting Information") should be lowercase.

⇒ We are sorry for our mistakes. We have corrected these in the Materials and Methods of revised manuscript.

Figure 1A: Is the dimerization due the HIS-tag? Is the effect observed caused by the monomer or the dimer?

⇒ Thanks for your kind and excellent comments. His-tags, due to their relatively small size (~2.5 kDa), are not believed to significantly interfere with the function and structure of a majority of proteins (Nat Methods. 2008 Feb;5(2):135-46; Curr Opin Biotechnol. 1992 Aug;3(4):363-9).

CB staining and Western blot analyses of whole-cell lysates using the anti-6x His tag antibody detected two protein bands with molecular weights of approximately 16 kDa and 36 kDa, which is consistent with a previous observation that hypoxic response protein 1 (Hrp1) migrated by SDS-PAGE as two major bands, corresponding to the monomer and dimer forms of the protein (J Microbiol Methods. 2013 Sep;94(3):192-8; J Med Microbiol. 2017 Jul;66(7):1033-1044).

Studies have shown that the hypoxia-induced dormancy response is established and maintained by the regulator DosR and ATP-related proteins such as Rv2626c (J Med Microbiol. 2017 Jul;66(7):1033-1044). Rv2626c/Hrp-1, is predicted to contain two cystathionine-b-synthase (CBS) domains (J Bacteriol 2001;183:2672–2676), which could potentially attach to a wide range of other protein domains, and possibly have regulatory roles in sensitizing proteins to adenosyl carrying ligands, or be involved in AMP/ATP binding in these proteins (J Mol Biol 2008;375:301–315; Biochemistry 2001;40:10625–10633). Because of that, I think it will be a dimer.

In Fig. 1F, his-tagged rRv2626c (dimer) was not detected and proinflammatory responses was decreased in cell lysates of TLR2^{-/-} BMDMs. Taken together, these results suggest that dimer form of

rRv2626c is internalized into the macrophages through TLR2, and TLR2/MyD88-dependent pathway is crucial for rRv2626c-mediated inflammatory response.

Figure 1B: To justify the term "BMDMs" more than one cell should be shown.

⇒ We are sorry for our mistakes. We have corrected these in the Fig. 1B and 5C in revised manuscript.

Figure 3: Have the THP1 cells been differentiated to macrophages or have these cells been used directly? If not, this should be specified in the manuscript.

⇒ We are sorry for our mistakes. We have added in the Materials and Methods of revised manuscript, as below.

Human monocytic THP-1 (ATCC TIB-202) cells were grown in RPMI 1640/glutamax supplemented with 10% FBS and treated with 20 nM PMA (Sigma-Aldrich) for 24 h to induce their differentiation into macrophage-like cells, followed by washing three times with PBS.

Figure 6G: The times following CLP in the schematic picture in the upper part of the Figure should be adapted to the times in the lower part of the picture.

⇒ We are sorry for our mistakes. We have corrected these in the Fig. 6G in revised manuscript.

13th Oct 2020

Dear Prof. Yang,

Thank you for the submission of your revised manuscript to EMBO Molecular Medicine. We have now received the enclosed reports from the referees that were asked to re-assess it. As you will see the reviewer is now globally supportive and I am pleased to inform you that we will be able to accept your manuscript pending the following final amendments:

1) Please revise your article for grammar and syntax (i.e. by an English native speaker) as suggested by the referee.

2) Please format manuscript sections according to the EMBO Molecular Medicine style. Please check "Author Guidelines" for more information.

<https://www.embopress.org/page/journal/17574684/authorguide#textformat>

***** Reviewer's comments *****

Referee #2 (Remarks for Author):

The manuscript has improved considerably. A good English proofreading may be worth considering.

Dear Prof. Yang,

Thank you for the submission of your revised manuscript to EMBO Molecular Medicine. We have now received the enclosed reports from the referees that were asked to re-assess it. As you will see the reviewer is now globally supportive and I am pleased to inform you that we will be able to accept your manuscript pending the following final amendments:

⇒ Thanks for your kind and excellent comments. We submit a revised version of our manuscript and a point-by-point response to the reviewers' comments. Detailed responses are described below.

1) Please revise your article for grammar and syntax (i.e. by an English native speaker) as suggested by the referee.

⇒ We did English-language editing of the article was carried out by Enago, branch of Crimson Interactive.

2) Please format manuscript sections according to the EMBO Molecular Medicine style. Please check "Author Guidelines" for more information. <https://www.embopress.org/page/journal/17574684/authorguide#textformat>

⇒ We have followed the Author Guidelines correctly.

***** Reviewer's comments *****

Referee #2 (Remarks for Author):

The manuscript has improved considerably. A good English proofreading may be worth considering.

⇒ Thanks for your kind and excellent comments. We did English-language editing of the article was carried out by Enago, branch of Crimson Interactive.

The authors performed the requested changes.

Corresponding Author Name: Chul-Su Yang

Manuscript Number: EMM-2020-12497-V3